

**Climate**
**of the Past**
Discussions

# Central Europe, 1531–1540 CE: The driest summer decade of the past five centuries?

Rudolf Brázdil[1,2], Petr Dobrovolný[1,2], Martin Bauch[3], Chantal Camenisch[4,5], Andrea Kiss[6,7],
Oldřich Kotyza[8], Piotr Oliński[9], Ladislava Řezníčková[1,2]

[1]Institute of Geography, Masaryk University, Brno, Czech Republic
[2]Global Change Research Institute, Czech Academy of Sciences, Brno, Czech Republic
[3]Leibniz Institute for the History and Culture of Eastern Europe (GWZO), Leipzig, Germany
[4]Oeschger Centre for Climate Change Research, University of Bern, Bern, Switzerland
[5]Institute of History, University of Bern, Bern, Switzerland
[6]Institute for Hydraulic Engineering and Water Resources Management, Vienna University of Technology, Vienna, Austria
[7]Department of Historical Auxiliary Sciences, Institute of History, University of Szeged,
Hungary
[8]Regional Museum, Litoměřice, Czech Republic
[9]Institute of History and Archival Sciences, University of Toruń, Poland

*Correspondence to*: Rudolf Brázdil (brazdil@sci.muni.cz)

**Abstract.** Based on three drought indices (SPI, SPEI, Z-index) reconstructed from the documentary evidence and instrumental records, the summers of 1531–1540 were identified as the driest summer decade during the 1501–2015 period in the Czech Lands. Based on documentary data, extended from the Czech scale to central Europe, dry patterns of various
intensities (represented, for example, by dry spells, low numbers of precipitation days, very low rivers and drying-out of water sources) occurred in 1532, 1534–1536, 1538 and particularly 1540, broken by wetter or normal patterns in 1531, 1533, 1537 and 1539. Information relevant to summer droughts extracted from documentary data in central Europe were confirmed in summer precipitation totals from a multi-proxy reconstruction for Europe
by Pauling et al. (2006) and further by self-calibrated summer PDSI reconstruction from tree-ring widths in OWDA by Cook et al. (2015). The summer patterns described are consistent with the distribution of sea-level pressure deviations from a modern reference period. Summer droughts were responsible for numerous negative impacts, such as bad harvests of certain crops, reduction and lack of water sources, and frequent forest fires, while in the
wetter summers central Europe was affected by floods. However, there are no indications of severe impacts of multi-country or multi-year effect. Reconstructions based on documentary data indicate that the summers of 1531–1540 constitute the driest summer decade in central Europe for the past five centuries, between 1501 and 2010 CE.

## 1 Introduction

Information related to droughts and their impacts derived from documentary evidence (Brázdil et al., 2018, 2019c) may be used for detailed analyses of individual severe drought events in the past (e.g. Munzar, 2004; Wetter et al., 2014; Kiss and Nikolić, 2015; Roggenkamp and Herget, 2015; Brázdil et al., 2016b, 2019a; Kiss, 2017, 2020; Bauch et al.,
2020; Camenisch et al., 2020) and for creation of a range of long-term drought chronologies (e.g. Martín-Vide and Barriendos Vallvé, 1995; Piervitali and Colacino, 2001; Domínguez-Castro et al., 2008; Diodato and Bellocchi, 2011; Brázdil et al., 2013a; Garnier, 2019; Tejedor et al., 2019; Przybylak et al., 2020). It is even possible, by means of documentary-based temperature and precipitation reconstructions, to calculate long-term series of drought indices





overlapping the pre-instrumental and instrumental periods (Brázdil et al., 2016a). Drought indices may also be reconstructed from drought-sensitive phenological series, such as those for grape harvest dates (Možný et al., 2016). Reported series of drought indices further allow comparison of the severity of drought episodes over several centuries (Brázdil et al., 2019b), as well as analysis of the effects of external forcings and large-scale climate variability modes upon droughts (Mikšovský et al., 2019).

Papers presenting overviews, both worldwide and European, of documentary-based drought studies have been published in recent years (Brázdil et al., 2018, 2019c). In Europe, the warm and dry weather of the year 1540 has attracted considerable attention. Wetter and Pfister (2013) considered 1540 as exceptionally hot for Europe, comparable with the outstanding summer event of 2003, when between 22,000 and 35,000 heat-related deaths were recorded across modern Europe (e.g. Schär and Jendritzky, 2004; Chase et al., 2006; Fischer et al., 2007). Subsequently, Wetter et al. (2014) presented 1540 as extremely dry on the broader European scale, coining the term "megadrought" for an unprecedented drought extending for 11 months. This paper and the term "megadrought" gave rise to controversy. Büntgen et al. (2015) questioned its results on dendroclimatological grounds, then Pfister et al. (2015) responded, pointing out the complexity of the whole drought phenomenon, as did the authors of the first 1540 drought paper. Subsequently, Orth et al. (2016) investigated whether European temperatures in 1540 are comparable with present-day (1966–2015) mean summer temperatures, while certain other authors using documentary data returned to the weather patterns of 1540 (Kiss, 2018; Pfister, 2018; Nowosad and Oliński, 2020).

However extremely dry 1540 proved to be, other summer drought episodes also occurred in the decade of 1531–1540, some of them close to one another, resulting in a notably high frequency of dry episodes in this decade within the long-term drought chronology of the Czech Lands dating back to 1501 CE (Brázdil et al., 2013a). Indeed, it may well have been the driest decade in half a millennium of reconstructed series of three summer drought indices (Brázdil et al., 2016a). Further, indications of dry summers during the 1531–1540 decade also follow from documentary data and related documentary-based hydroclimatic patterns of some other central European countries, such as Germany (Glaser, 2008) and Poland (Limanówka, 2001).

Previous findings lay behind this study and its aim of utilising and analysing documentary evidence from central Europe systematically to investigate the hypothesis that the summers of the 1531–1540 decade could have be the driest in over the past 500 years. Sect. 2 presents the documentary data used together with European precipitation, PDSI, sea-level pressure and temperature reconstructions. After descriptions of methods in Sect. 3, documentary data for the individual summers of 1531–1540 are reported, then related to spatial expression of precipitation totals and PDSI in Sect. 4, further complemented by circulation patterns of the summers investigated, long-term fluctuations of a number of climatological variables, impacts upon the life of the population, and people's responses to such summer weather. The discussion in Sect. 5 concentrates on the broader context of the results obtained as well as impacts and human responses. The last section summarises the results of the paper into few concluding remarks.

## 2 Data

### 2.1 Documentary data

Documentary evidence consists of a broad range of sources containing information about the weather, related phenomena, socio-economic consequences and human responses to them (Brázdil et al., 2005, 2010; White et al., 2018). The main types of documentary sources related to hydroclimatic patterns appear below, with examples from the summers of 1531–1540. Because many of the actual sources were dated according to the Julian calendar, 10



days have been added where appropriate to express them in the more recent Gregorian calendar (Friedrich, 1997):

**(i) narrative sources**

Different types of narrative sources contain descriptions of weather patterns and their impacts, including annals, chronicles, "books of memory" and private diaries. Large numbers of sources have already been professionally worked up and published as critical editions, but some still remain in archival collections in handwritten form. An example of subsequently published annals may be found in the records of Nikolaus Pol, a town scribe in Silesian Wrocław (see Fig. S1 in the Supplement) who wrote of summer 1532 (Büsching, 1819, p. 72): "*Dry summer. It did not rain for seven weeks. Cereals and pastures withered totally. There was no water in several settlements. In the countryside, it was impossible to grind* [grain]. *People had to go 10, 12, 18 miles* [converting the Breslau mile, *c*. 66, 79 and 119 km] *to* [get to] *mills. The River Oława dried up and had no water until St. Bartholomew's Day* [3 September]." Councillor scribes from Litoměřice in Bohemia reported patterns of the 1538 summer in their chronicle (Smetana, 1978, p. 133): "*It was very dry,* [people] *walked and rode* [by cart] *over the Elbe. The vintage started before Saint Jiljí* [11 September] *and exactly at* [the feast of] *Saints Peter and Paul* [9 July] the *grapes became soft; there had been no year like it for seventy years*."

**(ii) weather diaries**

More-or-less systematic daily records kept in weather diaries enable calculation of the frequency of certain characteristics (e.g. number of precipitation days) and the use of any related additional notes. In Polish Cracow, astronomers systematically entered qualitative daily weather records into their specialist calendars ("ephemerides"). Marcin Biem of Olkusz (*c*. 1470–1540), astronomer, theologian, and professor and rector of the Cracow Academy, kept daily records for 1531–1540, although from 1532 to 1534 they are incomplete or missing (Limanówka, 2001). For example, after some quite hot and nice days between 28 July and 2 August 1537, he described the weather in the first ten days of August (ibid., p. 30): "*3. Weather nice and hot, strong wind, rain in the night. 4. Cloudy in the morning, nice weather from noon, terrible thunderstorm with lightning in the evening. 5. Changeable weather, rain showers, lightning and wind. 6.–8. Rain every day. 9. Clear with light wind, more sunshine than rain. 10. Continuous rain and very strong wind.*" A somewhat unusual source consists of the daily military diaries of Turkish Emperor Suleiman, who described basic weather conditions from the central region of the Ottoman Empire, through the Balkan Peninsula to the Carpathian Basin and Lower Austria. For example, for the rainy end of June and beginning of July 1532 he noted (Thúry, 1893, p. 350): "*Year 938* [1532]. *month* [of] *Zil-kade* / […] / *17* [25 June]. *Friday. Stationed at Al-Kilise. Much rain fell.* / *18* [26 June]. *Saturday. Same place. It was raining.* / *19* [27 June]. *Sunday. Stationed at Elie* [*Ralie?*]. *Much rain fell.* / *20* [28 June]. *Monday. Stationed at Hisarlik. A great deal of rain* [fell]. / *21* [29 June]. *Tuesday. Arrived in Belgrade. A great deal of rain* [fell]. / *22* [30 June]. *Wednesday. Staying in place so that the army could cross the bridge.* / 23 [1 July]. *Thursday. The Padishah built a bridge* [opposite] *to Belgrade,* [we] *crossed the Sava bridge and settled in the Serim Plain. The Pasha came to meet him with an army of Rumelians. A great wind blew and it rained.* / *24* [2 July]. *Friday. Stayed in place. It rained. The Anatolian army crossed the bridge and stayed in Serim.* / *25* [3 July]. *Saturday. Stayed in place. It rained.* / *26* [4 July]. *Sunday. Stayed in place. It rained.* / *27* [5 July]. *Monday. Stayed in place. It rained.*"

**(iii) account books**

Books of accounts may contain records of municipal payments made for work related to weather and/or its extremes. In the town of Louny (north-western Bohemia), wages for municipal work in the previous week were paid regularly on a Saturday (Brázdil and Kotyza, 2000), providing indirect information about weather impacts. For example, records of harvest



works may indicate wetter patterns (interruption of harvest, late beginning and end of harvest) or drier ones (earlier and shorter harvest times). For example, the course of field labour in 1531 was evidently impaired by downpours in the week before 25 July when "*hay was pulled out of the water and dried*" and the harvesters were paid wages "*for three days, for half a day, not for the whole week*" (archival source AS14, fol. 148v). Other detailed account books helpful to the current study originate from Vienna (Austria) and Bratislava (Slovakia), where expenses related to grain and vine-harvesting work, or compensation for damage arising out of weather-related extremes, together with suggestions concerning the cause (and magnitude) of destruction, were recorded. Kiss (2018) analysed the grape and grain harvest dates from Vienna hospital accounts with respect to the extreme year of 1540. An example from the Bratislava accounts reports a late grape harvest, on 26 October 1533: "*On the day of Luca* [16 October] *the Chamberlain was asked to have the harvesting of the vineyards announced* […]" (AS1, K83/135, 176).

**(iv) chronograms**

Chronograms commemorate significant events or years in which people have been heavily affected. Selected letters (in capital letters or in bold type) are interpreted as Roman numbers indicating the year of an event. An example may be found in a commemoration of the 1540 drought in certain German sources (Riedel, 1862, p. 104):

"***EXICC**ata **L**e**VI**s **CV**r f**LVMI**na **C**er**V**i req**VIrI**s*" ["Why do you seek the swift deer when the rivers have gone dry?"], thus

$X + I + C + C + L + V + I + C + V + L + V + M + I + C + V + V + I + I = 10 + 1 + 100 + 100 + 50 + 5 + 1 + 100 + 5 + 50 + 5 + 1000 + 1 + 100 + 5 + 5 + 1 + 1 = 1540$).

The same sentence also appeared in handwritten remarks made concerning 1540 on calendars in southern Transylvania (Schuller, 1848, p. 356).

**(v) non-contemporaneous sources**

There exist considerable quantities of handwritten or printed materials describing a past event not directly experienced, but based on then-existing or lost documentary evidence, which may be characterised as documentary sources. However, they require a degree of critical work to establish their reliability and potential for complementing information derived from primary sources. An example may be found in a work by one Daniel Gomolcke, printed in 1737, which includes accounts, usually short, of weather conditions together with longer descriptions of the economic consequences for Silesian society, with additional remarks. For example, for summer 1534 he noted (Gomolcke, 1737, pp. 10–11): "*There was a great drought over Silesia in 1534, and a hot summer from Easter* [15 April] *to St. Bartholomew's day* [3 September]. *Water dried out,* [to the extent] *that watermills were out of operation, everything had to be ground by "Rossmühlen"* [mills powered by horses]*; high prices followed* […]." The Chronicle of Gaspar Hain from Levoča in Slovakia (Bal et al., 1910–1913), originating in the second half of the 17th century, is particularly reliable for the 16th century, since the author used official documentation from the local town archives, also providing precise names of the town mayors or notaries from whose notes he derived his information.

**(vi) weather compilations**

Weather compilations may contain information concerning various weather patterns and phenomena. However, they are often eclectic, gathered from a wide range of evidence that differentiates neither between primary and secondary accounts nor contemporaneous and non-contemporaneous sources. For example, weather data for 1531–1540 appears in several compilations related to central European countries such as Poland (Walawender, 1932; Rojecki et al., 1965) and historical Hungary (Réthly, 1962), but use of their data entails a critical evaluation against other existing reports. Some collections of historical-climatological data have been incorporated into generally accessible electronic databases, such as the Swiss



EuroClimhist (Pfister and Rohr, s.d.) and the German CLIMDAT (Militzer, 1998) and
Tambora (Riemann et al., 2015).

## 2.2 Hydroclimatic, pressure and temperature reconstructions

Three types of hydroclimatic reconstructions were used to document spatiotemporal
variability of summer (JJA) precipitation and droughts in central Europe:
(i) Brázdil et al. (2016a) calculated seasonal series of three drought indices for what is now
the Czech Republic in the 1501–2015 period: Standardised Precipitation Index – SPI (McKee
et al., 1993), Standardised Precipitation Evapotranspiration Index – SPEI (Vicente-Serrano et
al., 2010) and Z-index (Palmer, 1965). Reconstructed central European temperature series
(Dobrovolný et al., 2010) and Czech precipitation series (Dobrovolný et al., 2015), derived
from series of documentary-based temperature/precipitation indices and instrumental
measurements, provided the target data for these calculations.
(ii) Long series of instrumental precipitation measurements, documentary-based precipitation
indices and natural proxies sensitive to precipitation signals (tree-rings, ice cores, corals and
speleothems) were employed by Pauling et al. (2006) to calculate seasonal precipitation totals
throughout Europe for the 1500–2000 period. Their reconstruction combines gridded values
(0.5º latitude x 0.5º longitude) for the European land-mass (30ºW–40ºE, 30–71ºN) in the
years from 1500 to 1900 with a gridded reanalysis for the years 1901–2000 by Mitchell and
Jones (2005). The data are available at https://www.ncdc.noaa.gov/paleo-search/study/6342
(accessed on 8 January 2020).
(iii) Cook et al. (2015) used tree-ring widths to calculate gridded (0.5º x 0.5º) summer self-
calibrated Palmer Drought Severity Index (scPDSI) for *The Old World Drought Atlas*
(OWDA), covering the 0–2012 CE period (http://drought.memphis.edu/OWDA/Default.aspx;
accessed on 8 January 2020).
(iv) Circulation patterns for the summers of 1531–1540 in central Europe may be addressed
via maps of the European mean sea-level pressure (SLP) field over the Atlantic-European
sector based on 5° longitude x 5° latitude grids, as reconstructed by Luterbacher et al.
(2002b). Their data are available at https://www.ncdc.noaa.gov/paleo-search/study/6366
(accessed on 22 May 2020).
(v) Gridded European summer temperature data from the reconstruction by Luterbacher et al.
(2004) are similarly available at https://www.ncdc.noaa.gov/paleo-search/study/6288
(accessed on 22 May 2020).

## 3 Methods

With the territorial focus on central Europe, represented particularly by recent Germany,
Switzerland, Austria, the Czech Republic, Poland, Slovakia, Hungary and Transylvania
(western Romania), available documentary data from the various sources described in Sect.
2.1 were critically evaluated. The first step consisted of evaluation of the historical credibility
of the source. Primary sources were favoured, especially when the author was a direct
eyewitness to the event described or prepared a report from a short chronological perspective
or using older manuscripts. Secondary sources were used only after confirmatory comparison
with primary records or if, from the spatiotemporal viewpoint, they accorded with other
information. Historical place names for the locations to which the report or event described
was attributed were updated to their recent equivalents. Any archaic terminology within the
reports was translated into modern professional terminology. If a documentary source exists
only as a manuscript or separate print in an archive or library, it is cited as an archival source
(AS). If any of these archival documentary sources have already been the subjects of
professional attention and published, the corresponding reports are cited with respect to the
related publication rather than the original archival source.



Because the documentary dataset from central Europe is huge, only certain selected reports were used to describe the general character of summers in individual years during the 1531–1540 period in central Europe (Sect. 4.1). In order to characterise European spatial precipitation and drought patterns for each summer, the above documentary-based descriptions were considered in the light of the corresponding European maps of JJA precipitation totals after Pauling et al. (2006), expressed as percentage deviations from the 1961–1990 mean, and in terms of JJA scPDSI from OWDA after Cook et al. (2015), in which negative values describe dry patterns and positive wet (see Figs. 1–10). Mean SLP fields for individual summers in 1531–1540 were used to describe typical circulation patterns, together with their deviations from the 1961–1990 reference period over the Atlantic-European area (see Figs. 11–12).

To characterise JJA precipitation and drought patterns during the entire 1531–1540 decade, the corresponding gridded means of precipitation totals and scPDSI were calculated and expressed as deviations from the 1961–1990 reference period that indicated drier or wetter patterns (see Fig. 13). Precipitation and drought patterns were complemented by JJA temperatures taken from the European multi-proxy reconstruction by Luterbacher et al. (2004). To show the relationships of the above three variables to SLP patterns, their deviations from the 1961–1990 reference period were expressed with respect to SLP deviations (Fig. 14). Moreover, fluctuations in decadal means for the individual series investigated were expressed to put the decade analysed into the context of the last five centuries (see Fig. 15) and the 1531–1540 decade placed in order (from driest to wettest) within the entire series. This was based on series of JJA SPI, JJA SPEI and JJA Z-index for the Czech Republic (Brázdil et al., 2016a), on JJA precipitation totals in central Europe after Pauling et al. (2006), and on JJA scPDSI and JJA Drought Area Index after Cook et al. (2015). The central European region was demarcated by 5°–25°E longitude and 45°–55°N latitude. The Drought Area Index (DAI) was defined as the percentage area presenting values below a given threshold (Bhalme and Mooley, 1980). DAI was calculated on the basis of OWDA, using http://drought.memphis.edu/OWDA/Default.aspx (accessed on 8 January 2020) as an area with scPDSI less than or equal to –1. Further, the dates upon which grain and grape harvests began in the Czech Lands, Austria, Switzerland and France during 1531–1540 were included as proxy indicators of summer weather patterns (see Fig. 16).

## 4 Results
### 4.1 Weather and related events of the individual summers in 1531–1540
#### 4.1.1 Summer 1531 (Fig. 1)

In early May 1531, the River Elbe flooded in Saxony, although no rain was reported (Weck, 1679). A relatively cold period lasting until Pentecost (7 June) was reported from Mühlhausen in Thuringia (AS9) and other parts of central Germany (Spangenberg, 1572). In Bohemia, after a delay in the hay harvest in the Louny region, the course of field labour in 1531 was evidently impaired by downpours in the week before 25 July; harvesters were paid for only a few days, when they dried wet hay. The bad weather appears to have continued for the three weeks after 22 August, in which field labourers were paid no wages at all (AS14). Severe drought around 2 September was reported in Red Russia and south-eastern Poland, associated with the apparition of a comet, indicating droughts in August (Kronika Marcina Bielskiego, 1764). Summer weather conditions in Switzerland were somewhat unsettled, with rain, hail and thunderstorms in June, but there were also warm and sunny periods in July, then thunderstorms again in August (AS10). The grape harvest started on 5 October in the vineyards of the Vienna hospital in Austria (AS15). A good harvest of grain and grapes was mentioned for Litoměřice in Bohemia; the wine was described as "good" (Smetana, 1978). Extraordinary demand raised the barrel-price of wine in Litoměřice (ibid.). Plenty of wine,





again considered "good" was reported for Würzburg (Fries, 1713; Engel, 1950), Schweinfurt (Mühlich and Hahn, 1817), as well as for Kitzingen in Franken (Aldenberger, 1615) and Württemberg (Ginschopff, 1631), but also around Jena (Koch, 1928). Low wine prices are confirmed by a contemporaneous chronicler from Villingen (Roder, 1883).

5   In contrast, a bad harvest of grain and famine in Moravia are mentioned, without further specification (Steinbach, 1783), together with reports of a poor harvest (Dudík, 1868), while a severe shortage of grain is recorded by another source (Fischer, 1808). A severe shortage of grain was also mentioned in Silesian Wrocław by town scribe Nikolaus Pol (Büsching, 1819). High grain prices were reported for central Bohemia in the region around

10 Říp Hill (AS6) and for Halle and its surroundings in eastern Germany (Spangenberg, 1572; Dreyhaupt, 1749), as well as for Mühlhausen in Thuringia (AS9). The same problems were noted for Dortmund (Hansen 1887), Alsace and Swabia (Spangenberg, 1572), and particularly Villingen (Roder, 1883). A dearth of hay occurred in the area of Zurich (Switzerland) and grain was very expensive there (Hauser, 1905).

### 4.1.2 Summer 1532 (Fig. 2)

Martin Leupold von Löwenthal, the town scribe for Jihlava (Moravia), reported that a dry period occurred between 17 March and 3 July 1532, when the rain hardly dampened the dust (d'Elvert, 1861). Although four days of rain (18–21 July) at comparatively higher altitudes

20 ensured an adequate grain harvest, the dry spring and summer (ibid.) led to a bad harvest in both Bohemia (Tille, 1905) and Moravia (Dudík, 1868), for which Steinbach (1783) also mentions a dry year. In Trutnov (northern Bohemia), drought was recorded as so severe that fountains ran dry and fires broke out in the woods (Schlesinger, 1881). In the Louny region, the barley harvest was early, in the week before 25 June, although the wheat was harvested at

25 the usual time (AS14). Records kept by Pankraz Engelhart from Cheb in western Bohemia (Gradl, 1884) may also be attributed to 1532, describing the year as dry, with a dearth of grain, when people starved to death (dated erroneously to 1531). A similarly bad harvest was reported in Germany by the Heinrich Hug chronicle, but the wine was good (Roder, 1883). A source from the Brandenburg area of Germany (Riedel, 1862) mentions intense dryness for

30 many weeks; the soil was dry "knee deep". In Switzerland, the grain harvest was considered good but a plague of mice destroyed it (Blumer, 1853).

   A dry summer for 1532 was mentioned for Silesia, with no rain for seven weeks; grass and standing crops withered. There was such a shortage of water that water-mills could not operate and people had to travel 10 to 18 miles (i.e. from *c*. 66 to *c*. 119 km) to get their grain

35 milled. The River Oława was without water up to 3 September. There were also fires in Silesian towns, noted for 31 July in Wrocław and 4 September in Legnica (Büsching, 1819). A severe shortage of grain in Těšín was recorded by Biermann (1863).

   Gaspar Hain, in the Spiš area (Slovakia), recorded that there was no rain for 12 weeks between 3 April and 21 June: he mentions "great drought" in gardens, no grass in the

40 meadows, no pasture for animals, and that water mills could not work. The grape harvest in Hungarian Tokaj was early (Bal et al., 1910–1913). However, according to the daily reports of Emperor Suleiman, August, and probably July as well, could have been rainy in Hungary (Thúry, 1893).

45 ### 4.1.3 Summer 1533 (Fig. 3)

A chronicler for Bohemia records 13 floods before 3 July for Prague, and then another one (Zilynskyj, 1984). In the Louny region, the year was more or less average for work in the fields (Brázdil and Kotyza, 2000). In the week before 8 July, however, wages were paid to labourers who "*pulled hay out of the water*" (AS14, fol. 173v). It appears to have been rainy

50 in the week before 22 July as well, when wages were paid "*for turning wheat shocks*" and in



the two subsequent weeks with wages "*for spreading grain to dry and making new shocks*" (ibid., fol. 174r) and for "*spreading rye to dry*" (ibid., fol. 174v). Jan z Kunovic, in his intermittent daily records, notes rain on 24–26 and 28 July (Brázdil and Kotyza, 1996). The source from Jihlava records severe shortages despite a good harvest of grain (d'Elvert, 1861).

According to surviving data in the Vienna hospital accounts (AS15), the dates for the grain harvest (10 July), oat harvest (4 August) and grape harvest (10 October) were the latest of the decade. A wet June and July are mentioned in a report from Cracow (Poland) where, after persistent rains, the River Vistula flooded for the seventh time that year, between 17 July and 1 August (Maurer, 1878). The Silesian chronicler Michael Steinberg (Schönborn, 1878)

describes the summer as wet, with frequent thunderstorms and hailstorms that damaged the grain. The story is the same from Wrocław town scribe Nikolaus Pol who, in addition, mentions subsequent famine and a disastrous flood on 21–29 June on the River Oder (Büsching, 1819).

Southern German sources refer to 1533 in general as a "*cold and infertile year*"

(Ginschopff, 1631), they also report sour wine from the Breisgau and a lack of grain all over south-western Germany (Roder, 1883). Further afield, the water level of the River Rhine at Basel appeared to be high in March, April and June (Fouquet, 1999). Other Swiss sources describe periods of rainfall that damaged the grain in the fields, even leading to shortages (AS10; Blumer, 1853).

**4.1.4 Summer 1534 (Fig. 4)**

The barley harvest in the Louny region started very early in 1534, during the week before 23 June (AS14), but haymaking and the harvest of other cereals took place around the usual times (Brázdil and Kotyza, 2000). Harvest work finished early, in the week before 11 August.

The peas were picked almost immediately after this, and the hemp and millet harvested similarly early (ibid.). The summer proper was considered disastrously dry, with a lack of fodder for cattle and little water in the Elbe and Vltava rivers, but there was a good harvest of grain (Smetana, 1978; Kolár, 1987). After three months of drought, the water in the River Vltava was so low that people could walk across its bed (AS3). The fruit and grape harvests

were good and the year's wine judged "good" (d'Elvert, 1861; Smetana, 1978). The Vienna hospital accounts indicate that the grain harvest started on 5 July, oats on 30 July and grapes on 2 October (AS15). According to tithe payments, there was quite a good grape harvest in Sopron (Hungary) (AS2).

A severe lack of water, continuing until 1535, was reported in the Brandenburg area,

leading to high prices and shortages (Engel, 1598; Riedel, 1862). Hordes of caterpillars appeared in response to the drought. The weir at Jena could be crossed on foot (Koch, 1928). Low water levels and milling problems are known for Schweinfurt on the River Main (Mühlich and Hahn, 1817). *Mansfeldische Chronica* describes the summer as hot and very dry, due to which the grass in the fields and the leaves on the trees dried out. The lack of

water also made it difficult to find functioning water-mills to grind grain (Spangenberg, 1572). Dietrich Westhoff, a chronicler from Dortmund, reported an extremely hot summer and associated it with several city fires, including a blaze in Montabaur (Werlich, 1595; Hansen 1887). The connection between drought and fires in central Germany was also made by Spangenberg (1572). There were reports of a very warm summer in the Zurich area

(AS10).

A hot, dry summer in Silesia is also on record, the season lasting from 15 April until 3 September. Streams dried up, mills could not operate and famine followed (Gomolcke, 1737). This was confirmed by reports of the River Oder drying up and the use of horse- or hand-mills to grind grain (Kundmann, 1742). Grain sprouted on 25 April and there was a

good yield of cherries, but spells of great heat led to fires (Büsching, 1819). Similarly, the hot





summer presaged a lack of grain and fodder in Lusatia and water mills could not operate
(Roch, 1687; Gomolcke, 1737).

In contrast, Lesser Poland saw floods after some days of rain, occurring on 11–12 July
on the rivers Dunajec and Kamienna, as well as on the Vistula at Cracow (Kętrzyński, 1888).
South-western Germany reported a good grain harvest (Ginschopff, 1631) and floods on the
rivers Wertach and Lech south of Augsburg (Werlich, 1595). Flood damage was also reported
in central and southern Transylvania (Trauschenfels, 1860; Szilágyi, 1893). High prices
prevailed in Transylvania (Trausch, 1847; Gross and Seraphin, 1903a). Based on another
contemporary local chronicle, severe food shortage and hunger occurred in Sibiu (Vereins-
Ausschuß, 1851). Historical memories from the 17th-century indicate that this dearth lasted
for over three years (Barabás, 1880).

### 4.1.5 Summer 1535 (Fig. 5)
Bohemia was reported as hot and dry, with the streams drying up and frequent forest fires
(AS7). This pattern appears to be confirmed by field labour in the Louny region, where the
harvest started in the week leading up to 22 June, i.e. as early as in 1534. Haymaking took
place, as usual, in the following week. The rye harvest started in the week before 29 June and
was possibly the earliest for this crop in the entire 16th century. A very early start to the
harvest was also recorded for wheat (in the week before 13 July) and for oats (in the week
before 3 August). The harvest was over very early, in the week before 10 August (AS14). The
Vienna hospital accounts report the beginning of grain harvest as 2 July and the grape harvest
as 3 October (AS15).

In Silesia, the drought brought suffering to both people and livestock. Water mills
could not operate for ten weeks for sheer lack of water. The harvest of winter grain was
average and of summer grain, very bad (Büsching, 1819). A week of rainy spells starting on
August led to flood on the Vistula in Lesser Poland (Maurer, 1878). Brandenburg reported
an extremely hot summer (Riedel, 1862).

In Basel, according to the chronicle of Konrad Schnitt, the summer began with warm
and dry weather, but the weather changed in the middle of August and a rainy period
followed, lasting until November. Nonetheless, it was a productive year with very good grape
and grain harvests (Bernoulli, 1902). This is confirmed by historiography from around
Stuttgart (Ginschopff, 1631). In the Mansfelder Land area, west of Halle, harvests were
bountiful and grain prices plummeted to a fifth of what they had been in winter (Spangenberg,
1572). In Arnstadt, Thuringia, the grape harvest was plentiful (Olearius, 1701), as was the
grain harvest in Mühlhausen (AS9). However, much of this was spoiled in Jena by a plague of
caterpillars (Koch, 1937).

There are undated reports of floods in southern Transylvania to be found in the
contemporaneous chronicle of Hieronymus Ostermayer: the flooded River Olt damaged
Turnu Roșu Castle and the River Ghimbav did great damage in the Brașov area (Gross and
Seraphin, 1903b).

### 4.1.6 Summer 1536 (Fig. 6)
Summer 1536 was very dry in Bohemia and the water in the River Elbe was low.
Considerable damage was done by an eruption of caterpillars. Roses blossomed twice (AS12;
Smetana, 1978; Zilynskyj, 1984). Pankraz Engelhart and Andreas Baier (Gradl, 1884) speak
of a summer drought and heat in Cheb, together with frequent fires in forests and settlements.
Marek Bydžovský z Florentina (Kolár, 1987) mentions dry weather, with wells and streams
drying up and frequent fires from April to Christmas. In the Louny region, the timing of field
labour did not differ conspicuously from the average, with the exception of mowing the
aftermath, which was delayed to the week before 3 October (AS14). The Vienna hospital





accounts indicate that the grain harvest started on 2 July, oats on 3 August and grapes on
6 October (AS15). Although vineyards froze on 2 May around the town of Litoměřice
(AS12), there was "enough" good wine, whereas there was a poor grape harvest in Moravia
(Smetana, 1978). Ample good wine (see also Kolár, 1987), fruit and nuts, are mentioned by
Cheb chronicler Pankraz Engelhart, but also a dearth of grain after a poor harvest (Gradl,
1884).

     Several sources report a dry summer for Germany. Such reports come, for example,
from Meissen ("*an exceedingly dry summer*" – AS8, p. 188; confirmed for Dresden by Weck,
1679), Mansfeld (Spangenberg, 1572), Regensburg (Freiherrn von Oefele, 1878), Kitzingen
(Aldenberger, 1615) and Halle (Dreyhaupt, 1749). However, the grain harvest was good in
Saxony (Knauth, 1722). A hot summer was reported in Stuttgart, with dried-up wells and
brooks and a lack of water for people, livestock and milling (Ginschopff, 1631). Problems
with milling during the dry summer were reported in Erfurt and the municipality decided to
provide *Rossmühle*, i.e. mills run by horses (Falckenstein, 1738). In Jena, a single day of rain
in May was worth reporting, since it was the first of the year (Koch, 1937). Sources in
Pomerania and Silesia reported frequent thunderstorms with torrential rains and related
damage, particularly in June–July (Gomolcke, 1737; Hubatsch and Arnold, 1968).

     Basel chroniclers report a prevalence of very warm and dry patterns during the
summer. The vintage was therefore very good and the wine prices low. However, there was a
dearth of hay and lack of fodder for the cows led to a shortage of butter (Vischer and Stern,
1872; Bernoulli, 1902). The chronicler Hans Salat described very dry weather conditions in
central Switzerland, where water mills could not operate for a long time for lack of water,
although an abundant harvest was mentioned for the region (Baechtold, 1876).

### 4.1.7 Summer 1537 (Fig. 7)

Wet patterns in central Germany started with several days of rain and subsequent flooding of
the River Saale at Jena, starting on 9 May (Koch, 1937). Rain and flooding continued
throughout May (Roch, 1687; Müller, 1806; Koch, 1937). In Bohemia, floods after three days
of rain were reported from 30 May to 1 June. These culminated with considerable damage on
June (AS5; Zilynskyj, 1984). Similarly, a flood was recorded on 30–31 May on the Lusatian
Neiße (Nysa), when it did considerable damage to the town of Görlitz in Saxony (Müller,
1806; Struve, 1870) and other places (Rojecki et al., 1965). The summer of 1537 was
described as wet in Varnsdorf (Bohemia), and the grain grew well (Palme, 1913). This
concurs with similar records for Silesia. There are further confirmations of a wet summer,
with reports of periods of continuous rain and flooding for June and July from Wrocław
(Büsching, 1819). In the Louny region, the beginning of work in the fields took place at the
usual time, with the exception of the hemp harvest (Brázdil and Kotyza, 2000). The start of
haymaking in the week before 3 July was evidently affected by rain, because "*when raking
the hay, everything was soaked with water*" (AS14, fol. 223v). The haymaking and the harvest
were interrupted in the week before 7 August, apparently due to rain, because no wages were
paid for field labour during that time (AS14). The Vienna hospital accounts indicate the dates
upon which harvests started: grain on 6 July, oats on 29 July and grapes on 2 October (AS15).

     A parallel finding consists of reports of many thunderstorms during summer 1537 in
Saxony (Roch, 1687). For central Germany, *Mansfelder Chronik* reports continuous rain in
June (Spangenberg, 1572), and further flooding in the town of Görlitz in Saxony (Müller,
1806) and in Jena (Koch, 1937). At the end of June, it was so cold that people started to use
their ovens for heating (AS9; Spangenberg, 1572). Andreas Baier in Cheb described the
whole year of 1537 as wet (Gradl, 1884). In central Switzerland, after a rainy May, the wheat
and oat harvests were good, as were those of cherries and apples (Baechtold, 1876). A rainy
June with flooding and a cold summer are mentioned in German sources for Mansfeld





(Spangenberg, 1572), as well as for Regensburg (Freiherrn von Oefele, 1878) and for Jena (Koch, 1937). Floods were also reported in the north of Germany (Lappenberg, 1861). A good harvest of crops and fruit was observed around Stuttgart (Ginschopff, 1631).

### 4.1.8 Summer 1538 (Fig. 8)
A hot, very dry summer occurred in Bohemia, with forest fires in places (AS7; Gradl, 1884), and in the Erzgebirge mountain range as well (Arnold, 1812). The dry weather evidently led to a dearth of grain mentioned by Johann Mathesius (AS4). The water in the River Elbe was so low that the bed could be crossed on foot and by cart. In the Louny region, the harvest started early, in the week before 25 June (AS14). The wheat harvest began in the week before 9 July, the earliest recorded for this crop in the 16th century. The peas also ripened very quickly indeed, and were already being gathered in the week before 23 July; accelerated in similar fashion, the oat harvest began in the following week and ended in the week before 13 August (AS14). Grapes started to soften on 9 July and the vintage began before 11 September (Smetana, 1978). Ample good wine was reported (Rezek, 1879). The Vienna hospital accounts indicate harvest dates for grain starting on 6 July and grapes on 5 October (AS15).

The summer in Dortmund (Germany) was described as hot, with a negative effect on crops (Hansen, 1887). Michael Steinberg in Silesia mentions spells of heat before 19 June and an early harvest, on 3 July (Schönborn, 1878). In eastern Pomerania there was "*nearly no water*" in the River Vistula (Henneberger, 1595; Curicke, 1688). It was a productive year in Transylvania and prices were low (Gross and Seraphin, 1903b; Armbruster, 1984). This was probably the case in other parts of the Carpathian Basin as well (Horváth, 1868; Weber, 1894).

### 4.1.9 Summer 1539 (Fig. 9)
The chronicler Pavel Mikšovic recorded a flood at Louny after a cloudburst on 24 July (AS13). In the same region, much of the harvest came in very early: the haymaking, the general harvest, the rye, oats, hemp and millet (Brázdil and Kotyza, 2000). In addition to satisfactory harvests of grain, vegetables and fruit, narrative sources also mention ample good wine (d'Elvert, 1861; Smetana, 1978). A Silesian source refers to a pleasant, dry harvest (Büsching, 1819). Vienna hospital accounts indicate harvest dates starting on 5 July for grain, 24 July for oats and 4 October for grapes (AS15).

From the intermittent daily weather records kept by Jan z Kunovic, there is a note for 23 August ("*rain up to this point*") and heavy rain fell on 26 August (Brázdil and Kotyza, 1996). This spell of rain may have been responsible in part for five large floods on the River Saale recorded for Jena (Koch, 1928, 1937). One of these had already occurred on 6 August (Koch, 1937). Only a day later, on 7 August, there was a major flood in Meißen, Thuringia and Franconia (Aldenberger, 1615). Summer 1539 in Meißen was marked by a dearth of grain (Weck, 1679), and a similar situation prevailed around the Hartz mountains (Spangenberg, 1572) and, again, in Jena (Koch, 1937). For Erfurt, the Falckenstein (1738) chronicle reports sour wine because of a wet autumn and too little sunshine in summer. Intense heat occurred in Jena, around 2 July (Koch, 1937). Farther west, in Trier, the grape harvest was reportedly abundant, as was the wine from it (Zenz, 1962). The same is known of Nürnberg (Aldenberger, 1615). This contrasts strongly with information from Dortmund (Hansen, 1887) that indicates a dearth of grain and outright famine there. Ginschopff's chronicle (1631) reports good harvests after an average summer. In Switzerland as well, the grain and the grape harvests were good (AS10; Baechtold, 1876). The Ostermayer chronicle mentions relatively low cereal and wine prices in this year for Transylvania (Gross and Seraphin, 1903b).



### 4.1.10 Summer 1540 (Fig. 10)

Documentary sources in Bohemia speak above all of a hot, dry summer, shortages of water, and forest fires (AS4; AS7; AS13; Dudík, 1858; d'Elvert, 1861; Kolár, 1987). In the surroundings of Jihlava, a dry period with bad yields of vegetables and beet is recorded from
6 April until the end of the year. The harvest started before 9 July and grain crops were cut before 11 August (d'Elvert, 1861). Very dry and warm conditions with severe shortages of grain and vegetables occupied much the same time period in Uherský Brod (Zemek, 2004). Pavel Mikšovic (AS13) relates that a hot, dry period lasted from 26 May to 13 October with rain only on 8 August, and with warm weather until the end of the year. In Cheb, Pankraz
Engelhart (Gradl, 1884) speaks of a dry year, in which sowing and work in the fields went right on until the end of the year. Andreas Baier (ibid.) mentions a dry summer and warm weather until the end of the year. For the Louny region, the very earliest starts to the harvest were recorded: the beginning and the end of the harvest (in the weeks before 22 June and 3 August, respectively), the barley harvest (the same as for the general harvest), oats (in the
week before 20 July) and peas (in the week before 13 July) (AS14). The grain harvest is described as "poor" (Gradl, 1884; Kolár, 1987) through "medium" (AS7) up to "ample" but with poor yields of other crops (Rezek, 1879). 1540 was an excellent year for wine in Bohemia, with an abundant harvest of grapes (AS7; AS11; d'Elvert, 1861).

    Severe heat and drought in summer and autumn afflicted Silesia, where there was
practically no rain for six months. Many streams dried up and the water in the River Oder turned green. There were frequent forest fires and livestock suffered from hunger and thirst (Büsching, 1819). Similarly, severe heat, forest fires, poor harvest, shortages and famine were mentioned for Bohemia, Silesia and Lusatia (Gomolcke, 1737). In Greater Poland, summer and autumn were also very dry; it did not rain until the beginning of winter. Rivers were
exceedingly low, brooks, ponds and wells dried up and the land was desiccated to dust (Rojecki et al., 1965). A hot summer with dried-up ponds, and cattle needing to be moved great distances for water, as well as poor growth of cereals were reported for Pomerania and eastern Prussia (Hubatsch and Arnold, 1968).

    *Gastainerische Chronica* (Rohr, 2007) mentions warm, dry weather from the end of
March to mid-August, with a very bad grain harvest and dried-up springs and brooks in the Salzburg area of Austria; forest fires were reported for Carinthia. The Vienna hospital accounts indicate quite an early start to the grape harvest, on 25 September, and the oats harvest beginning on 29 July. The grain harvest which started on the particularly early date of 26 June was the earliest of all the documented grain harvests of the century (AS15; Kiss
2018). Fabricius (AS8) confirms a very warm and dry summer in Meissen; there was plenty of wine but a lack of garden produce. A report from the Brandenburg area mentions a very hot summer, during which forests were on fire in many places and waterways dried up, but the wine was good (Riedel, 1862).

    Several Swiss sources remark upon a notably hot, dry year. In the Lucerne area, the
dry weather led to many forest fires. Water levels were low and some fountains even dried up (Baechtold, 1876). In Basel, according to chronicler Fridolin Ryff, it did not rain more than three times between the beginning of summer and November and therefore many water mills could no longer operate. Moreover, water levels were low everywhere; it was even possible to ride or walk across the River Rhine. The wine, fruit and grain harvests were abundant
(Vischer and Stern, 1872). In Zurich, the heat lasted from the end of February until mid-September and not more than six days of rain occurred during this period. The water levels were so low that mills could not operate, and water for livestock had to be carried great distances in barrels. Nonetheless, the year was productive (Egli, 1904). In the north-east of Switzerland, the soil looked burnt by the drought and wells as well as fountains dried up.



Mills could not operate either, while trees suffered great damage due to lack of water
(Baechtold, 1906).

  A contemporaneous calendar inscription from Transylvania suggests that the solar
eclipse on 17 April 1540 was followed by great heat, when many springs dried up, while in
other places forest fires occurred (Schuller, 1848). Writing at the time, the historian Ambrus
Somogyi, among others, also referred to a hot midsummer period (Simigianus, 1800). In the
town accounts for Brașov, fires were mentioned twice, in summer and autumn (Rechnungen,
1889).

**4.2 Circulation patterns in the summers of 1531–1540**
SLP maps may be used as indicators of possible circulation patterns in the decade studied.
The mean JJA SLP field in the Atlantic-European area during the 1961–1990 reference
period, together with deviations for the individual summers of 1531–1540 with respect to this
reference, appear in Fig. 11. Mean JJA SLP field in the reference period shows the strong
Azores high south-west of Europe with its ridge of high pressure extending to central Europe
(Fig. 11a). Low pressure is especially evident in the south-eastern Mediterranean and adjacent
land area. In terms of SLP deviations in the individual summers of 1531–1540 (Fig. 11b),
drier summers in central Europe correspond with pressure increases over the European land-
mass and decreases over the Atlantic Ocean. The cores of pressure increases are especially
marked over south-west Scandinavia (1534, 1536 and 1540) and extend from there to central
Europe (1532 and 1535). In the dry summer of 1538, positive increases run from the south-
west to the north-east (with the core region extending from the British Isles to south-western
Scandinavia), with one arm continuing to Scandinavia and a second arm reaching central and
south-eastern Europe. In somewhat wetter ("normal") summers, marked pressure decreases
make up a dominant belt extending from the British Isles to central and south-eastern Europe
(1531, 1533, 1537). SLP for summer 1539 indicates quite positive NAO patterns: pressure
decreases south of Iceland and increases south-west of the Iberian Peninsula.

  In general, the mean patterns for summer 1531–1540 (Fig. 12a) are similar to those for
mean JJA SLP field in the reference period (Fig. 11a). Their deviations with respect to 1961–
1990, however, are more interesting (Fig. 12b). The border between negative and positive
pressure deviations tends to extend from the north-west (east of Iceland) to the south-east
(central Mediterranean) and divides the Atlantic-European area into two parts. The decreases
in SLP are located west of this border and the majority of them are statistically significant
(particularly in the central part of the Atlantic and in the Iberian Peninsula), while pressure
increases occur towards the east, again at their highest in Scandinavia and lower in central and
southern Europe.

  While SLP fields explain the exceptional character of the 1531–1540 summers clearly,
the summer index of the North Atlantic Oscillation (NAO), reconstructed by Luterbacher et
al. (2002a), achieves only slightly positive values (not shown). They show no significant
deviation from the previous or subsequent decades; in fact, they tend to decrease from
positive values at the beginning of the 16th century to negative thereafter.

**4.3 Hydroclimatic variability of the 1531–1540 summers in longer-term context**
The spatial precipitation, drought and air temperature patterns of the 1531–1540 summers,
with respect to the summers in the 1961–1990 reference period in central Europe, appear in
Fig. 13. Lower JJA precipitation totals (Pauling et al., 2006) tended to concentrate towards the
south-eastern part of central Europe (eastern Bohemia, Moravia, southern Poland, eastern
Austria, Slovakia and Hungary). While in western central Europe negative differences with
respect to the reference period were much smaller, in the northern part of Poland this decade
was identified as rather wetter (Fig. 13a). Turning to JJA scPDSI according to OWDA (Cook





et al., 2015), spatial patterns in these terms contrast quite strongly with precipitation. Drier patterns with respect to the reference period were reconstructed for much of Germany (north-west, south-west, east), the Swiss Plateau and Bohemia. Much wetter conditions were reconstructed for Austria, the greater part of Poland, Slovakia and Hungary (Fig. 13b).

Decadal central European fields of the two hydroclimatic characteristics show only very weak parallels, with the Pearson correlation coefficient at only 0.08, thus statistically insignificant. Because PDSI depends not only on precipitation, but also on temperatures, Fig. 13c shows anomalies of mean summer temperatures compared to the same reference. Although this decade was 0.8°C warmer than the 1961–1990 period, the degree of warming varied over

central Europe. Germany together with western Bohemia, and a smaller south-to-north belt over Hungary, Slovakia and Poland, were notably warmer. The smallest positive temperature difference extended from northern Italy to Austria and much of Poland was also characterised by less marked differences.

Comparing the above three variables (precipitation, scPDSI and temperature) with

deviations in SLP over central Europe (Fig. 14), a clear significant negative correlation in precipitation and a significant positive correlation for air temperature are evident. Overall, the positive correlation between SLP and scPDSI is distinctly weaker, especially during the years 1531 and 1536. This is partly due to the fact that, although drought is closely related to both precipitation and temperature, it also depends strongly upon their temporal distribution and

the persistency of the periods with low precipitation and high temperatures.

Based on summer drought indices (SPI, SPEI, Z-index) for the Czech Republic (Brázdil et al., 2016a), calculated from documentary-based temperature and precipitation reconstructions for 1501–2015, JJA in the 1531–1540 decade was the driest in the past 510 years (Fig. 15a). Drought patterns in 1721–1730 nearly matched those of this decade. A

similar pattern holds for gridded JJA precipitation totals in central Europe (the 5°–25°E and 45°–55°N window), reconstructed by Pauling et al. (2006) for the 1501–2000 period, followed by the 1701–1710 decade (Fig. 15b). The uniqueness of the 1531–1540 decade is weakened when decadal characteristics in the 1501–2012 period for the same window are calculated from OWDA (Cook et al., 2015). The 1531–1540 decade in terms of scPDSI for

JJA emerges as the ninth driest, while JJA DAI is the eighth at a threshold of –1 (Fig. 15c,d).

### 4.4 Impacts of the 1531–1540 summers and societal responses to them

The documentary sources for central Europe in 1531–1540 are of especial value for their reports of the impacts of weather patterns on the beginning, duration and quantity of harvests,

particularly those of grain and grapes. Certain records even enable the tracking of continuous series of harvest beginnings for the Czech Lands, Austria and Switzerland (Fig. 16a). In general, it may be anticipated that preceding warmer and drier patterns contribute to earlier harvest dates and *vice versa*. Czech and Swiss series of grape-harvest dates (Fig. 16b) reflect this scheme very well, clearly expressed in the early starts for harvesting in 1540. On the

other hand, the Vienna hospital accounts relating to the grape harvest express, in comparison, a lower inter-annual variability and only partial agreement. Similar tendencies are also apparent for the grain harvest in Dobroměřice in north-west Bohemia (Brázdil and Kotyza, 2000) and for the winter rye harvest in northern Switzerland/south-west Germany (Wetter and Pfister, 2011), except for relatively long delays in harvest dates, particularly in 1536 and, to a

lesser degree, 1537 (Fig. 16a). The Vienna series, with smoothed fluctuations, agrees in part with anticipated trends; more differences appear in series of the winter wheat harvest days in the Czech Lands (Možný et al., 2012).

Weather patterns for summers, together with their previous winters and springs, were reflected in good/bad yields of basic crops, influencing food sources in ways that could result

in rising prices and the onset of starvation, even famine. For example, Michael Steinberg



(Schönborn, 1878) published the prices of wheat and rye in his chronicle for Silesia, 1532–1541, unfortunately without further information. Wheat was the most expensive in the wet, cold year of 1533 with prices at 21–28 Groschen (Gr.), then in 1534 (18–24 Gr.) and in 1538 (16–24 Gr.). Rye also commanded its highest price there (20–25 Gr.) in 1533, although he

reported 25 Gr. for rye in 1534 and 1539). Similarly, Nicolas Pol (Büsching, 1819) noted a peak in prices in Wrocław for 1533: 25–26 Gr. for wheat and 22–24 Gr. for rye. However, the highest prices of all occurred in 1540: 26 Gr. for both wheat and rye. Steinberg in Silesia noted only 16–23 Gr. for wheat in 1540, with no information for rye (Schönborn, 1878).

      Severe shortages and very high prices for cereals were reported for 1535 in

Transylvania (Schuller, 1848; Vereins-Ausschuß, 1851). According to *Album Oltardianum*, an early 17th-century chronicle, and also Wolfgang Bethlen, a history writer at around the same time, Transylvania suffered from famine in that year (Bethlen, 1782; Trauschenfels, 1860). So great was the hunger that people of both sexes and all ages lost their minds, walking around almost naked and consuming "unclean things". Bethlen also mentioned

cannibalism. Thousands of people starved to death. Corpses could be encountered on the streets, their mouths full of grass (Bethlen, 1782). In Făgăraş, desperate poor people turned to eating dead dogs and cats (Trauschenfels, 1860). A century later, the history writers Wolfgang Bethlen and Matthias Miles considered this three-year period of hunger in historical Hungary, especially severe in its northern parts (today's Slovakia) and Transylvania, almost equal to the

greatest known famine of the 16th–17th-centuries, which occurred in 1603 (Bethlen, 1782; Armbruster, 1984). Food availability and prices could be also negatively influenced by plague as was reported, for example, in 1531 in Hungary (Vereins-Ausschuß, 1851), or by other epidemic diseases, as cited for the same year in Swiss Winterthur (AS10).

      Warm, dry summers gave rise to particular problems in central Europe. When streams

and rivers fell or dried out, it became difficult or impossible to grind grain in regions that relied upon water mills. For example, the dry summer of 1536 forced the Erfurt municipality to consider the creation of a *Rossmühle*, a mill powered by horses (Falckenstein, 1738). Watering cattle was also very difficult, a problem specifically mentioned in relation to the droughts of 1536 in Germany (Ginschopff, 1631) and 1540 in Silesia (Büsching, 1819).

However, it should also be pointed out that people had no problems with access to water in the towns situated near big rivers such as the River Vistula in Poland, even in times of extreme drought. On the other hand, wet summers were sometimes accompanied by floods that could cause material damage and flooding of agricultural land, or even loss of human lives, as was apparent in 1533 in Bohemia (Zilynskyj, 1984), Poland (including Silesia)

(Büsching, 1819; Maurer, 1878), Switzerland (Fouquet, 1999) and Transylvania (Vereins-Ausschuß*,* 1851; Gross and Seraphin, 1903b), and again in 1537 in Bohemia (Zilynskyj, 1984), Silesia (Büsching, 1819) and Germany (e.g., Spangenberg, 1572; Lappenberg, 1861; Freihern von Oefele, 1878; Koch, 1937).

      In times of extended and severe droughts, the number of fires in towns and forests

rose. For example, forest fires were reported in 1532 in Silesia (Büsching, 1819), in 1536 and 1538 in Bohemia (AS7; Gradl, 1884; Kolár, 1987), and in 1540 in the Brandenburg area (Riedel, 1862), Carinthia (Rohr, 2007), Silesia (Büsching, 1819), Bohemia, Silesia and Lusatia (Gomolcke, 1737), and Transylvania (Schuller, 1848).

**5 Discussion**
**5.1 The summers of 1531–1540 in European context**
The character of the summer weather described in Sect. 4.1 and 4.3 may be confirmed by other central European studies. For example, Limanówka (2001) used qualitative daily weather records from ephemerides in Cracow (Poland), covering the first half of the 16th

century, to classify temperature-precipitation patterns. She interpreted the summers of 1531–



1540 as: 1531, warm, mean precipitation (37 precipitation days); 1532–1534, no data; 1535, cool, mean precipitation (42); 1536, moderate, dry (35); 1537, moderate, humid (47); 1538, warm, dry (39); 1539, moderate, humid (42); and 1540, very hot, extremely dry (13).

   A range of documentary evidence has been used to interpret the summer patterns of
1531–1540 in Germany (Glaser, 2008) and in Switzerland (Pfister, 1988, 1999). A very warm, dry summer in 1532 (particularly August) was recorded in southern and eastern Germany (Glaser, 2008), while a warm June and August were derived for Switzerland (Pfister, 1988). Pfister (1988) interprets a wet summer for 1533 in Switzerland, while Glaser (2008) gives a cooler and wetter summer in Germany. A warm, dry summer for 1534 follows
from the interpretations of documentary data for both Germany (Glaser, 2008) and Switzerland (Pfister, 1988). Glaser's analysis (2008) for 1535 in Germany provides spatially different intensities of heat in the summer, which became moist towards the end. In Switzerland (data missing for July), the months of June and August were interpreted as average and June, further, as wet (Pfister, 1988). The analysis for 1536 in Germany gives a
mild, dry spring, a very warm and very dry summer, and a warm, very dry autumn (Glaser, 2008). For Switzerland, Pfister (1988) mentions warm months from May to November, and dryness from July to October (July–September were classified as anomalously warm-dry – Pfister, 1999). Glaser's analysis (2008) for 1537 gives a cold and very wet summer for Germany, particularly in June. While the data for summer 1538 in Switzerland are incomplete
(Pfister, 1988), Glaser (2008) interprets warm and dry summer for Germany. Glaser (2008) speaks of summer 1539 as wet, although with less rain in western and eastern Germany. Pfister (1988) mentions a warm June and August for Switzerland but the data for July are missing. The extraordinary extremity of the year 1540 is recorded by Pfister (1988) for Switzerland, who mentions a sequence of warm months from April to December and dry ones
from February to October. Similar reports of warm and dry patterns also come from Germany (Glaser, 2008).

   Based on the quantitative precipitation reconstruction from documentary sources for the Czech Lands by Dobrovolný et al. (2015), summer precipitation totals for the 1530s were lower than in the 1961–1990 reference period, and also compared with the preceding (1520s)
and subsequent (1540s) decades. Further, low precipitation in the 1530s in central Europe was confirmed in four proxy reconstructions based on tree-ring widths, although they refer to slightly different seasons (see Fig. 9 in Dobrovolný et al., 2018 for further details).

   High temperatures in addition to low precipitation could also have contributed significantly to droughts. For example, summer temperatures for central Europe reconstructed
from documentary sources (Dobrovolný et al., 2010) indicate that the 1531–1540 decade was 0.9°C warmer than the 1961–1990 reference period. The former decade was significantly warmer (at a significance level of 0.05) than the previous (1521–1530) and subsequent (1541–1550) decades. Moreover, a majority of the 15 various proxy reconstructions from central Europe performed by Brázdil et al. (2013b) have confirmed the exceptionally warm character
of the 1530s.

   It follows from Figs. 1–10 that summer weather patterns in western Europe were similar to those in central Europe. In particular, the summers of 1534, 1536, 1538 and 1540 were dry not only in central Europe, but the area affected by drought extended over the greater part of western Europe. Pribyl and Cornes (2020) mentioned repeated warm and dry
episodes in the 1530s and a drought in 1540 in England. In the Netherlands, Alsace and France, summer 1532 was sunny and very dry with a good grape harvest (Bournon 1895; Buisman, 1998). Summer 1533 in the Netherlands was wet with prolific rain in June, July and August. As a result, the water in several rivers was high, leading to flooding and burst dikes in the Nijmegen, Doesburg and Den Bosch areas (Buisman, 1998). Summer 1534 in the Paris
district was hot and very dry from April until autumn and the grain harvest was good





(Bournon, 1895). On the other hand, summer 1535 was rainy and quite cold in the Paris area and the Alsace (Buisman, 1998). Summer 1538 was dry in the Netherlands (ibid.). In the Netherlands and in Alsace, summer 1539 was quite warm with some dry spells, but there were periods of rain. The hay and grape harvests were good in both areas (ibid.). Dry and warm patterns for 1540 in western Europe have been confirmed by Wetter et al. (2014). In the Netherlands, this summer was extraordinarily hot. People had not experienced anything similar for hundreds of years. Fountains dried up and even large rivers such as the Rhine, Maas, Leie and Scheldt ran very low. There was almost no rain at all. Wildfires and town fires broke out in several places. Harvests were very early, and while damage to vegetation was reported in some places, in others the harvests were abundant (Buisman, 1998).

Only sparse information concerning drought episodes in the summers of 1531–1540 is available from eastern Europe. In contrast to central and western Europe in 1533, severe drought was reported in Russian annals for the East-European Plain. In its central part (the Moscow area) the annalist wrote: "*No rain from June to September but drought* [zasucha] *and heavy fog* [generated] *by hot smoke from burnt-out forests and dried-up peat bogs. The sun at three* [in the afternoon] *was red and it was impossible to see by it.*" (Shmakin et al. 2013, p. 53). The situation was similar in the northern part of the plain (the Novgorod area): extended spell without rain, dried-up bodies of water, forest fires, smoke (ibid.).

**5.2 Impacts and societal responses to the summers of 1531–1540**

Partial comparisons may be drawn between the impacts of drought of drought and society's responses to it in central Europe through further similar documentation. For example, in the Paris region (Bournon, 1895), after a cold and rainy May 1531, the market price of grain was very high the following summer. A very good vintage was reported in 1532, with an abundance of wine at market. Hot, dry weather also occurred in this area in the months of May, June and July 1534, when a lack of hay and oats was mentioned. However, it appears that the wheat harvest, was at least sufficient. In Burgundy and Alsace, the grape, fruit and grain harvests were early and good in 1536 (Buisman, 1998). French grape-harvest days at Beaune (Labbé et al., 2019) correlate closely with the general character of the summers of 1531–1540, as well as with that of other central European series (Fig. 16b).

Turning to the consequences of dry weather, central European documentary sources often describe problems with the availability of water and activities related to its levels. For example, during the extremely hot, dry summer of 1540 in the Netherlands, the water levels in rivers were so low that people could cross substantial rivers such as the Lys, the Scheldt, the Meuse and the Rhine with "dry feet" (Descamps, 1852).

Soon after the 1531–1540 decade examined herein, in 1542, locusts invaded central Europe (Brázdil et al., 2014). However, Count Lestár Gyulaffy, a Transylvanian diplomat, mentioned in his historical annals, completed in 1605, that the infamous central-European locust invasion of the early/mid-1540s had already begun in 1539 in Transylvania. Although he was not an immediately contemporary author (he was born in 1557), his dating of such an early outbreak is worthy of mention, as his work otherwise contains a correct historical chronology of political and other events (Szilágyi, 1893).

**6 Conclusion**

The results of this analysis of summer hydroclimatic patterns during the 1531–1540 decade in central Europe may be summarised as follows:
(i) Reconstructions based on documentary data indicate that the summers of 1531–1540 were the driest summer decade in central Europe of the past five centuries, i.e. following 1501 CE. Dry patterns were expressed at their clearest in 1532, 1534–1536, 1538 and particularly in



1540, which was extremely dry. Reconstructions based on tree-ring widths tend to attribute the JJA scPDSI of 1531–1540 to the end of the first ten, the driest, summers.

(ii) Drier summers in central Europe correspond with pressure increases over Fennoscandia and central Europe and to statistically significant pressure decreases in the central part of the Atlantic Ocean and south-west Europe. Wetter summers correspond with pressure decreases extending from the British Isles to central Europe.

(iii) The impacts of summer droughts within the 1531–1540 period were reflected in familiar manifestations, such as bad harvests of certain crops, reduction or complete lack of water sources, frequent fires in towns and forests, contrasting with floods and all their negative effects in wetter summers. However, there are no indications that the more severe impacts were of multi-country or multi-year natures.

(iv) Despite the fact that precipitation totals and PDSI represent different hydroclimatic characteristics, the multi-proxy precipitation reconstruction by Pauling et al. (2006) and scPDSI tree-ring width reconstruction by Cook et al. (2015) indicate only little agreement in the spatial distribution of drier and wetter patterns over (central) Europe in the summers of 1531–1540.

(v) The current study demonstrates the high potential of documentary data in the detailed analysis of weather patterns of past summers in Europe. These occurred in times of only natural processes. This is very relevant to the evaluation of recent and future droughts arising out of the effects of both natural and anthropogenic forcings.

**Data availability.** The series of hydroclimatic, pressure and temperature reconstructions were obtained from following databases: http://drought.memphis.edu/OWDA/Default.aspx for scPDSI, https://www.ncdc.noaa.gov/paleo-search/study/6342 for precipitation, https://www.ncdc.noaa.gov/paleo-search/study/6366 for sea level pressure, and https://www.ncdc.noaa.gov/paleo-search/study/6288 for air temperature. Sources of weather and related information were extracted from quoted archival sources and references.

**Author contributions.** RB designed and wrote the paper with contributions from all co-authors. PD analysed spatial distribution and did analyses of precipitation, PDSI, SLP and temperature in European and central European scale. Following colleagues contributed with documentary data: MB for Germany, CC for Switzerland and western Europe, AK for former Hungary and Transylvania, OK for the Czech Lands and PO for Poland. LŘ worked with Tambora and EuroClimhist databases and finalised the figures. All authors have read and comment the last version of the manuscript.

**Competing interests.** The authors declare that they have no conflict of interest.

**Special issue statement.** This article is part of the special issue "Droughts over centuries: what can documentary evidence tell us about drought variability, severity and human responses?". It is not associated with a conference.

**Acknowledgements.** RB, PD and LŘ were supported by the Ministry of Education, Youth and Sports of the Czech Republic for the SustES – Adaptation strategies for sustainable ecosystem services and food security under adverse environmental conditions project, ref. CZ.02.1.01/0.0/0.0/16_019/0000797. MB was supported by a Freigeist Fellowship from the Volkswagen Foundation. AK acknowledges financial support from the Austrian Science Fund, project ref. I 3174. This study was undertaken by CRIAS, a working group of the Past Global Changes (PAGES) project, which in turn received support from the Swiss Academy of Sciences and the Chinese Academy of Sciences. The authors also acknowledge Climate



Explorer (https://climexp.knmi.nl), used to plot the maps that appear as Figs. 1–10. Tony
Long (Dalmellington) helped work up the English.

**Archival sources**

[AS1] Archiv hlavneho mesta SR Bratislavy, Magistrát mesta Bratislavy (AMB-A/XXIV.1):
Komarna kniha/Kammerbuch: K1–56 (1434–1500).
[AS2] Győr-Moson-Sopron Megye Soproni Levéltára IV. 1003. Acta decimalia.
[AS3] Moravská zemská knihovna Brno, sign. 25363: Kniha Duchovní o velikých skutcích
Pána Boha všemohoucího: Rozličnými hystoriemi starými i novými ozdobená. V níž se
obsahuje vysvětlení: Mohouli Čarodějníci a Čarodějnice sami od sebe Povětří, Kroupy,
Bouře, Hromobití vzbuditi a vyvésti. Z Písem Svatých, Učitelův Křesťanských i Pohanských
sepsaná a nyní v nově vydaná. Od kněze Jana Štelcara Želetavského z Želetavy, toho času
Faráře v Městečku Mnichovicích. Létha Páně: MDLXXXVIII.
[AS4] Moravská zemská knihovna Brno, sign. L.VIII.187: Chronica der kayserlichen freyen
Bergstadt Sanct Jo[a]chimsthal der zuvor die Conradsgrün genent war. Anhang in Johann
Mathesius, Sarepta oder Bergpostill. Sampt der Jo[a]chimsztalischen kurzen Chroniken.
Psalm. CXLVIII. Berg und Thallober den HLRRN. Nürnberg, MDLII.
[AS5] Moravský zemský archiv Brno, fond G 21 Sbírka starých tisků, inv. č. 526, sign.
III/160: Rerum Boemicarum Ephemeris sive Kalendarium historicum. Ex reconditis veterum
annalium monumentis crutum. Authore M. Procopio Lupacio [de] Hlavaczov aeo, Pragensi.
Opus nunc primum in lucem editum, una cum coronide ac locuplere personarum ac rerum
memorabilium indice. In idem kalendarium. Eruta dum patriae monumenta Lupacius edit, et
bonus est civis, doctus et historicus. Suscitat, amplificat, manifestat, promovet, ornat,
maiores, patriam, tempora, gesta, duces. M. Bern. Sturmii. Pragae, Anno M.D.LXXXIIII.
[AS6] Národní knihovna České republiky Praha, sign. XVII. B. 14: Urbář kláštera
doksanského založený r. 1522 s přepisy listin právních zboží téhož kláštera z 16. století (do r.
1588) a se zprávami kronikářskými z let 1453–1531.
[AS7] Regionální muzeum Litoměřice, inv. č. SV 14142: Kalendář Historický. To jest krátké
poznamenání všech dnuov jednoho každého měsíce přes celý rok. K nim přidány jsou některé
paměti hodné Historie o rozličných příhodách a proměnách, jak národuov jiných a zemí v
Světě, tak také a obzvláštně národu i Království Českého z hodnověrných Kronik. S pilnosti
sebráno, vytištěno a vydáno prací a nákladem M. Daniele Adama z Veleslavína. Vytlačeno v
Starém Městě Pražském. Leta posledního věku: MDXC.
[AS8] Sächsische Landesbibliothek – Staats- und Universitätsbibliothek Dresden, sign.
VW/93/135: Georgii Fabricii Chemnicensis Rerum Misnicarum libri VII. Electorum Saxoniae
lib. I. Marchionum Misnensium lib. I. Annalium urbis Misnae libri III. Siffridi Misnensis
presbyteri Epitomes lib. II. Omnia nunc recens edita. Lipsiae. Curante Ernesto Voegelino.
Cum provilegio.
[AS9] Stadtarchiv Mühlhausen, sign. 61: Anonymus, Die Mühlhäusischen Alterthümer in
einer Chronica vorgestellet zum Nützlichen Gebrauch vor die Nachkommen aus vielen Alten
Schrifften und Chronicken zusammengetragen Im Jahr Christi 1800, Zweyther Theil.
Illustrierte Sammelhandschrift aus dem Besitz von Amalia Mohts, geb. Becke.
[AS10] Stadtbibliothek Winterthur: Goldschmid, J. J.: Erzellung seltsammer Natur-
Geschichten ungwohnten Jahrgängen: theuren und wohlfeilen Zeiten, Sterbensläuffen und
anderen derglichen Sachen so sich bey uns in der Statt Winterthur zuogetragen haben.
[AS11] Státní okresní archiv Litoměřice, fond Archiv města Litoměřice, sign. IV B 1a: Kniha
pamětní litoměřických městských písařů 1570–1607.
[AS12] Státní okresní archiv Litoměřice, fond Archiv města Litoměřice, st. sign. 12:
Letopisecké záznamy v litoměřickém právním rukopise ze 14. stol. označený nově "Das
Magdeburger Recht".



[AS13] Státní okresní archiv Louny, fond Archiv města Louny – kroniky, sign. Ch 1: Chronica civitatis Launensis in Boemia auctore Paulo Mikssowicz servo consulari.
[AS14] Státní okresní archiv Louny, fond Archiv města Louny, sign. I E 10: Iste liber pro hospitali et curia allodiali deputatus inicium sui cepit post exustam hanc urbem etc.
[AS15] Wiener Stadt- und Landesarchiv, MA 8. 1.7.1.1. B 11 – Spitalmeisteramtsrechnung 1531, 1533–1540.

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



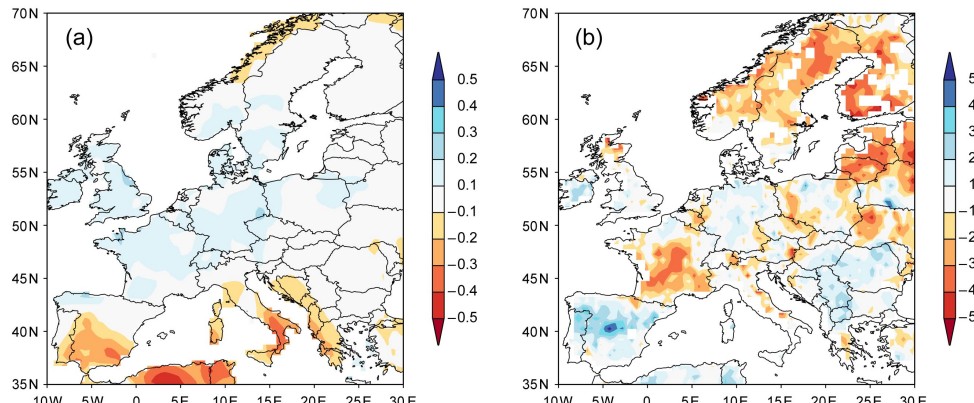

**Figure 1.** (a) 1531 JJA precipitation totals in Europe expressed as percentage deviations (x100) from the 1961–1990 mean (Pauling et al., 2006); (b) JJA scPDSI for 1531 in Europe according to OWDA (Cook et al., 2015).



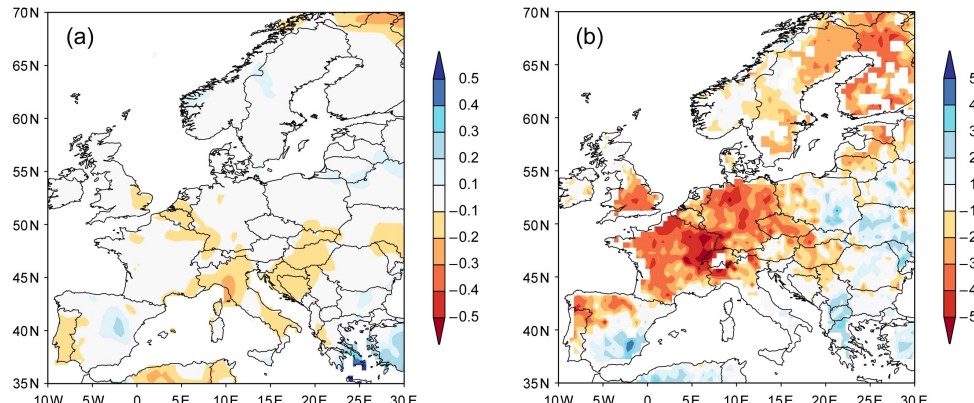

**Figure 2.** (a) 1532 JJA precipitation totals in Europe expressed as percentage deviations (x100) from the 1961–1990 mean (Pauling et al., 2006); (b) JJA scPDSI for 1532 in Europe according to OWDA (Cook et al., 2015).



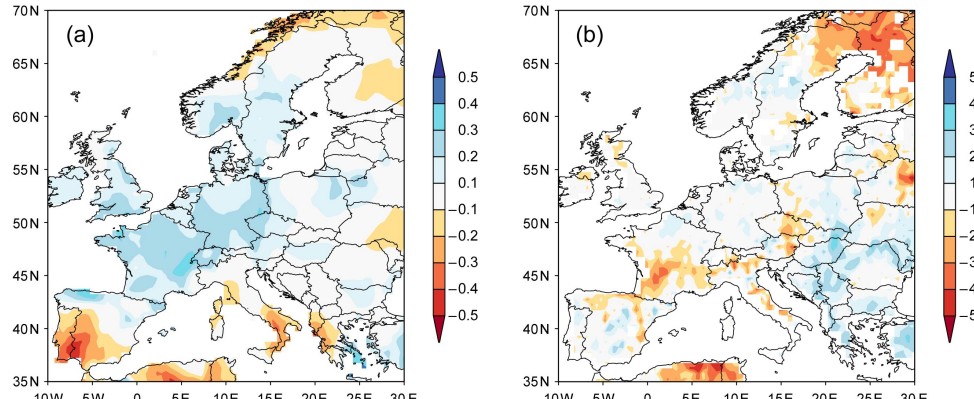

**Figure 3.** (a) 1533 JJA precipitation totals in Europe expressed as percentage deviations (x100) from the 1961–1990 mean (Pauling et al., 2006); (b) JJA scPDSI for 1533 in Europe according to OWDA (Cook et al., 2015).



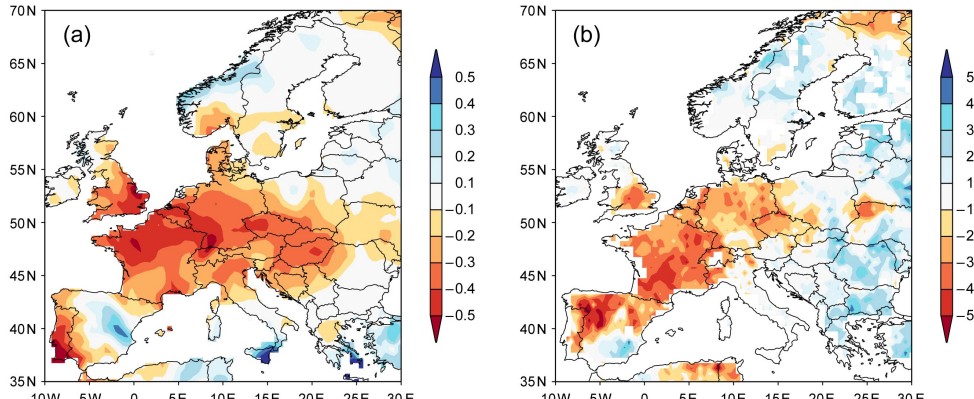

**Figure 4.** (a) 1534 JJA precipitation totals in Europe expressed as percentage deviations (x100) from the 1961–1990 mean (Pauling et al., 2006); (b) JJA scPDSI for 1534 in Europe according to OWDA (Cook et al., 2015).



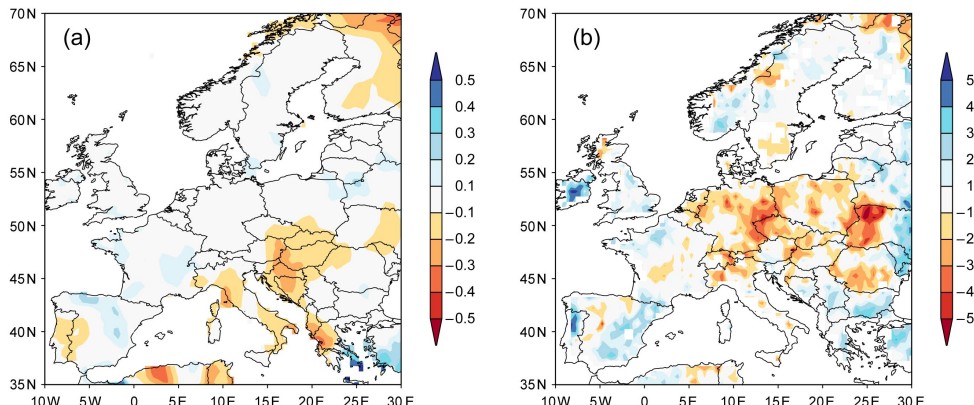

**Figure 5.** (a) 1535 JJA precipitation totals in Europe expressed as percentage deviations (x100) from the 1961–1990 mean (Pauling et al., 2006); (b) JJA scPDSI for 1535 in Europe according to OWDA (Cook et al., 2015).



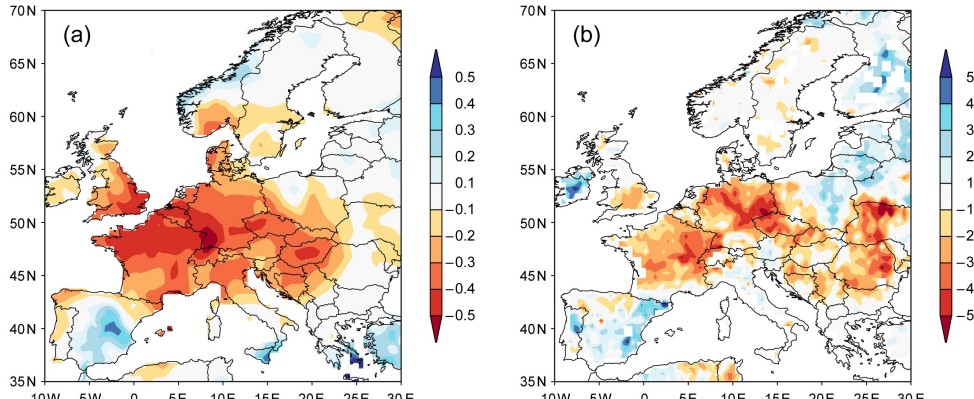

**Figure 6.** (a) 1536 JJA precipitation totals in Europe expressed as percentage deviations (x100) from the 1961–1990 mean (Pauling et al., 2006); (b) JJA scPDSI for 1536 in Europe according to OWDA (Cook et al., 2015).



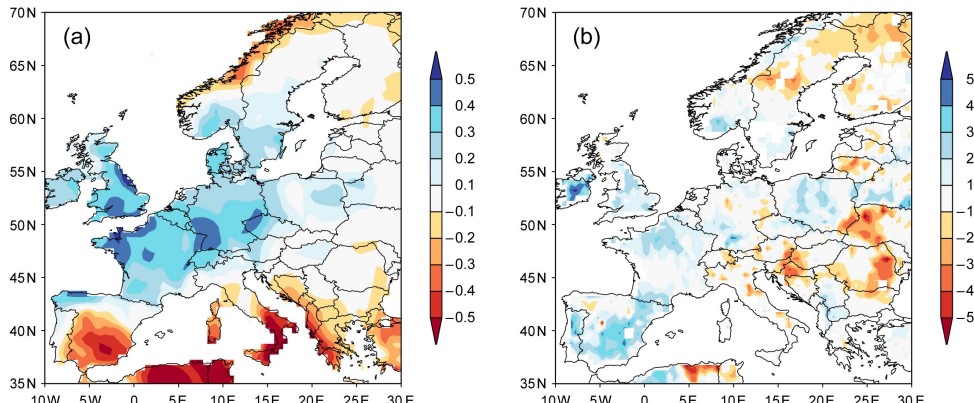

**Figure 7.** (a) 1537 JJA precipitation totals in Europe expressed as percentage deviations (x100) from the 1961–1990 mean (Pauling et al., 2006); (b) JJA scPDSI for 1537 in Europe according to OWDA (Cook et al., 2015).





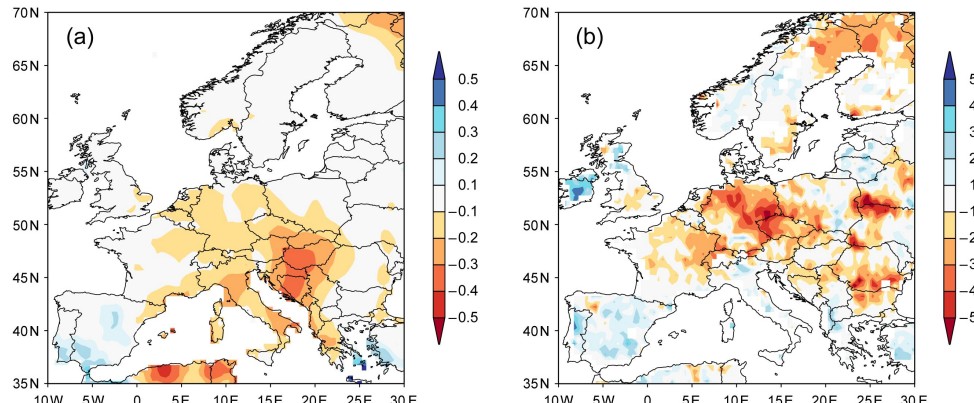

**Figure 8.** (a) 1538 JJA precipitation totals in Europe expressed as percentage deviations
(x100) from the 1961–1990 mean (Pauling et al., 2006); (b) JJA scPDSI for 1538 in Europe
according to OWDA (Cook et al., 2015).





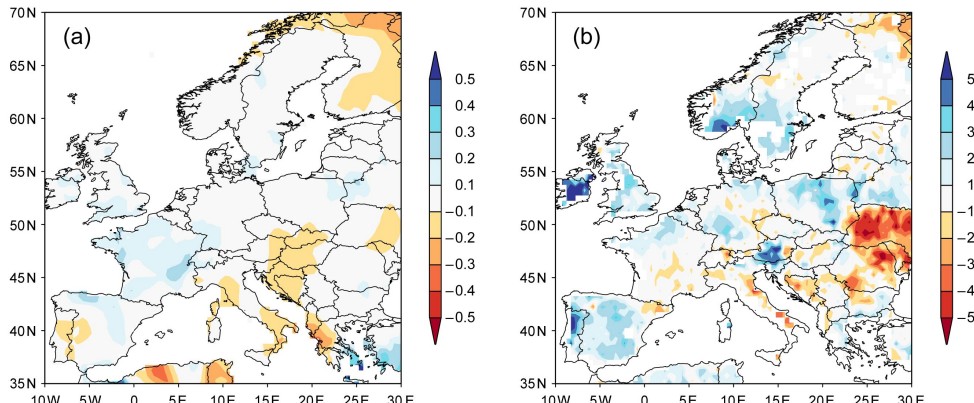

**Figure 9.** (a) 1539 JJA precipitation totals in Europe expressed as percentage deviations (x100) from the 1961–1990 mean (Pauling et al., 2006); (b) JJA scPDSI for 1539 in Europe according to OWDA (Cook et al., 2015).





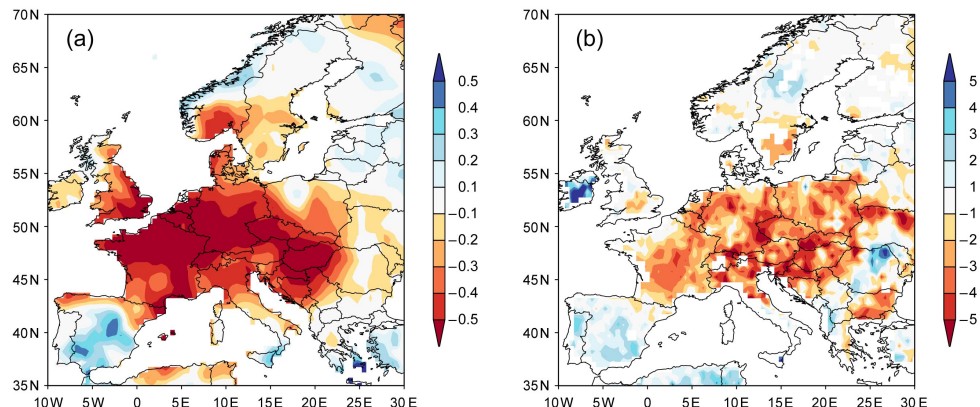

**Figure 10.** (a) 1540 JJA precipitation totals in Europe expressed as percentage deviations (x100) from the 1961–1990 mean (Pauling et al., 2006); (b) JJA scPDSI for 1540 in Europe according to OWDA (Cook et al., 2015).

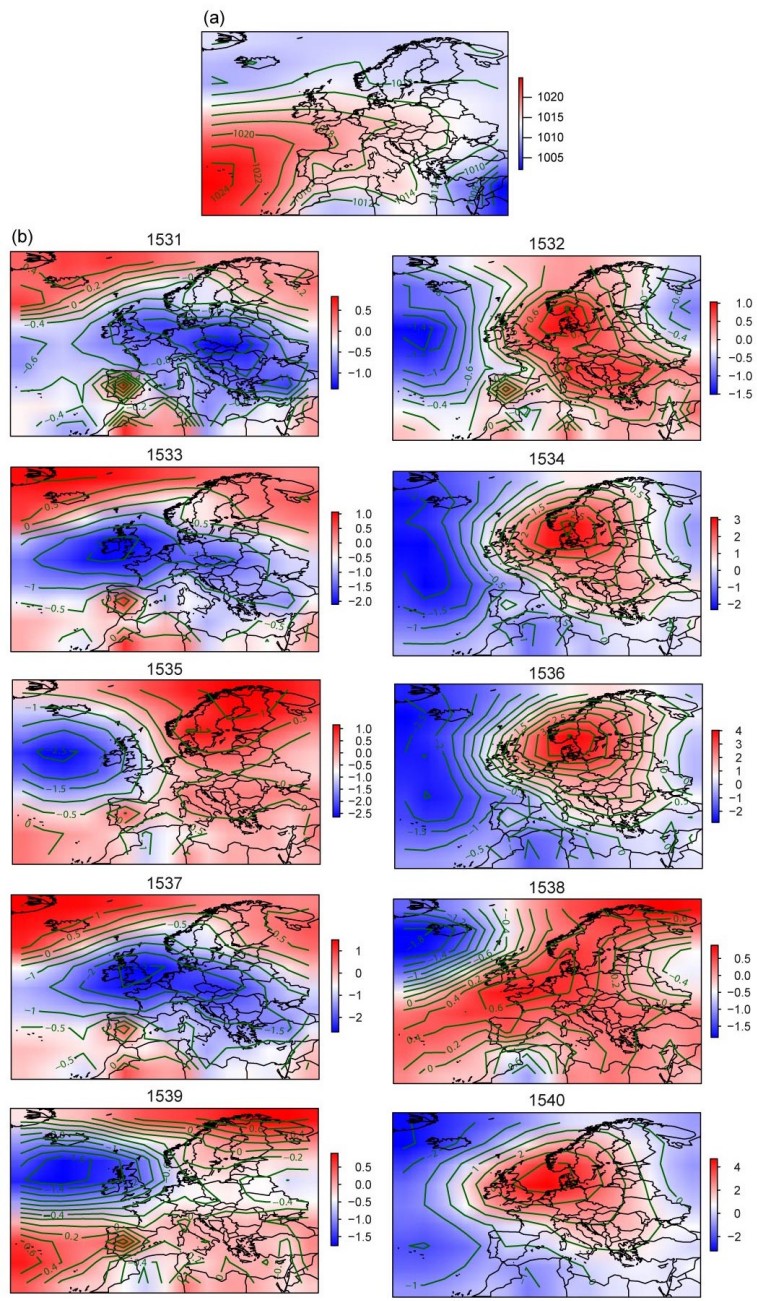

**Figure 11.** (a) Mean JJA SLP field in the Atlantic-European area during the 1961–1990 reference period and (b) anomalies in the mean SLP fields of the individual summers in 1531–1540, based on data from Luterbacher et al. (2002b) (available at https://www.ncdc.noaa.gov/paleo-search/study/6366, accessed on 23 May 2020).



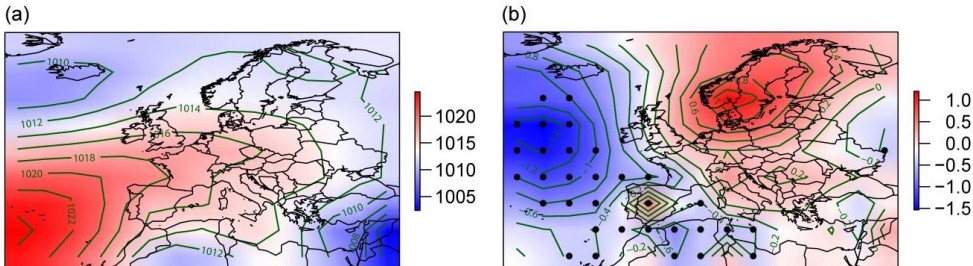

**Figure 12.** Mean JJA SLP field for 1531–1540 (a) and deviations from the 1961–1990 reference period (b) in the Atlantic-European area. Points indicate grids with statistically significant deviations at a significance level of 0.05.



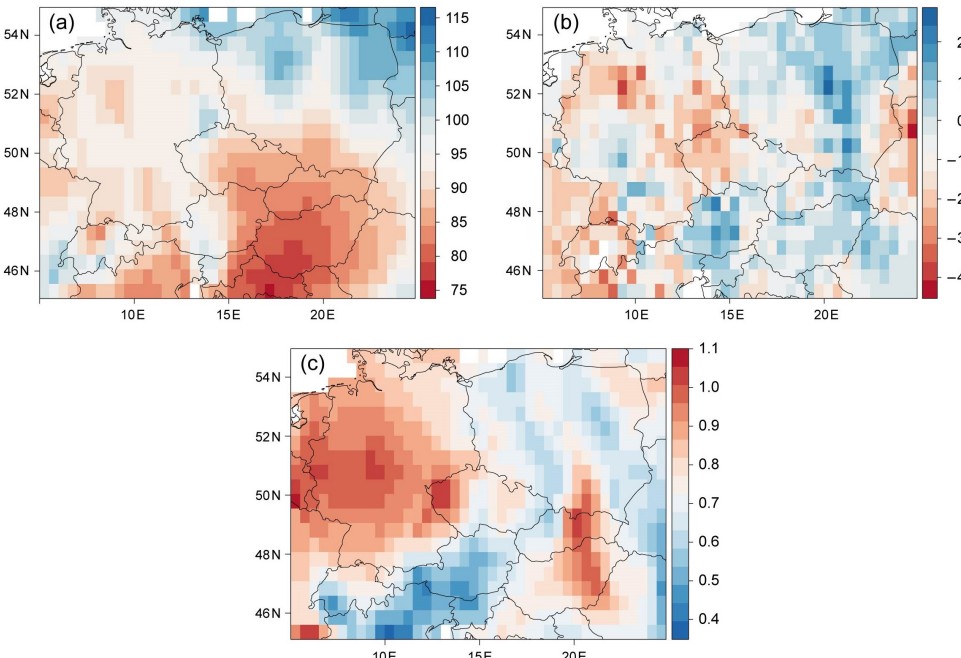

**Figure 13.** (a) Mean 1531–1540 JJA precipitation totals expressed as percentages of the 1961–1990 mean (Pauling et al., 2006), (b) mean 1531–1540 JJA scPDSI according to OWDA (Cook et al., 2015) expressed as deviations from the 1961–1990 mean, and (c) mean 1531–1540 JJA temperatures according to Luterbacher et al. (2004) expressed as deviations (°C) from the 1961–1990 mean in the central European region.



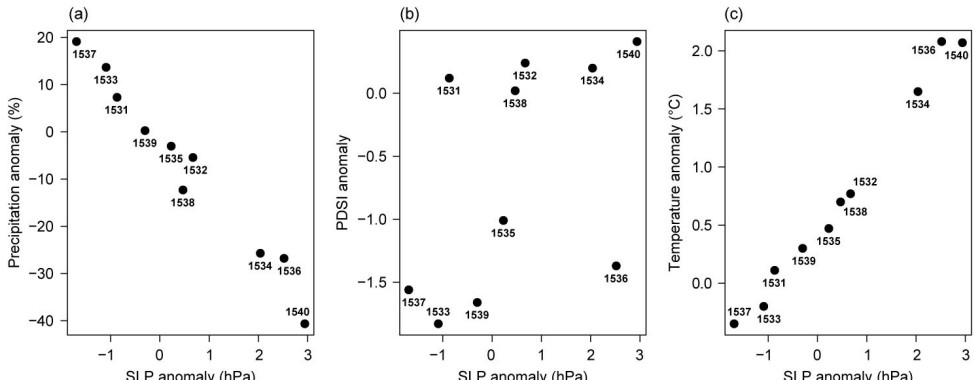

**Figure 14.** Relationship between mean SLP anomalies (Luterbacher et al., 2002b) and mean precipitation (a), PDSI (b), and temperature (c) anomalies for the individual summers of 1531–1540 in central Europe (the area demarcated by 45°–55°N and 5°–25°E). Data sources: (a) Pauling et al. (2006), (b) Cook et al. (2015), (c) Luterbacher et al. (2004).

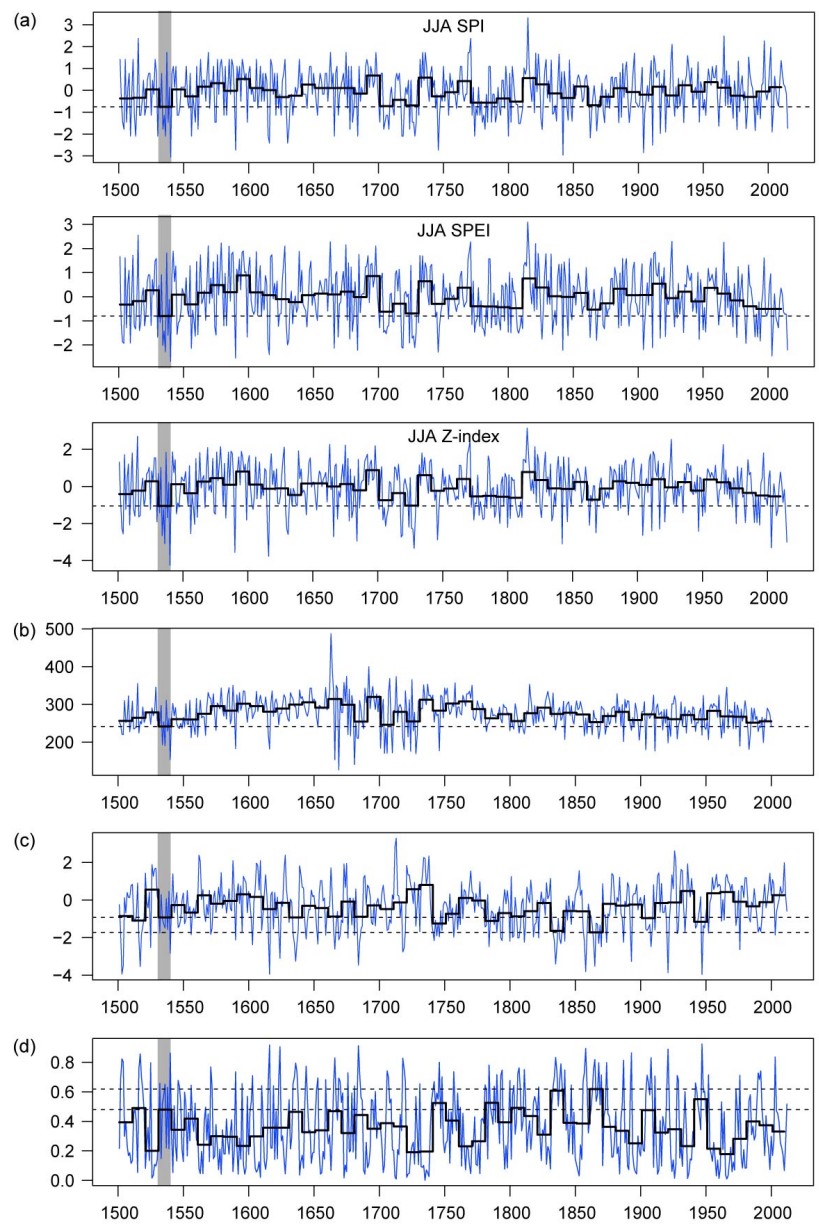

**Figure 15.** Decadal fluctuations in hydroclimatic characteristics: (a) JJA SPI, JJA SPEI and JJA Z-index for the Czech Republic, 1501–2015 (Brázdil et al., 2016a); (b) JJA precipitation totals within the 5°–25°E and 45°–55°N window, 1501–2000 (Pauling et al., 2006); (c) JJA scPDSI of the 5°–25°E and 45°–55°N window, 1501–2012 according to OWDA (Cook et al., 2015); (d) JJA DAI (threshold –1) of the 5°–25°E and 45°–55°N window, 1501–2012 according to OWDA (http://drought.memphis.edu/OWDA/Default.aspx, accessed on 8 January 2020). The grey band indicates the 1531–1540 period.





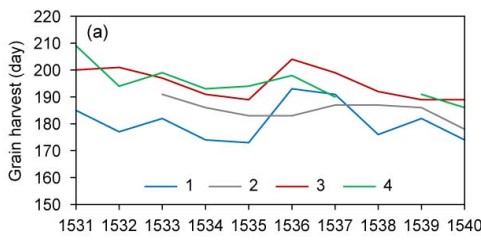 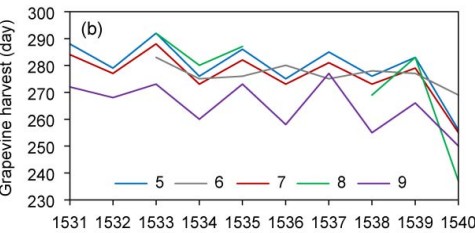

**Figure 16.** Harvest starting days, expressed as days after 1 January, for selected locations and regions in central Europe and France in 1531–1540: (a) grain (1 – Dobroměřice: Brázdil and Kotyza, 2000; 2 – Vienna: AS15; 3 – winter wheat – Czech Lands: Možný et al., 2012; 4 – winter rye – northern Switzerland/south-western Germany: Wetter and Pfister, 2011); (b) grapes (5 – the Louny region: Brázdil et al., 2019d; 6 – Vienna: AS15; 7 – Bohemian viticulture region: Možný et al., 2016; 8 – Switzerland: Meier et al., 2007; 9 – Beaune, France: Labbé et al., 2019).