# Peer review of "Central Europe, 1531–1540 CE: The driest summer decade of the past five centuries?"

_Climate of the Past, 2020_

## Referee Comment (RC1) · Anonymous Referee #1 · 23 Aug 2020

General comments:

This is an excellent paper that brings together a range of data sources and datasets in order to analyse the characteristics and significance of the 1530s, which the authors have previously shown to be the driest summer decade of the period 1501-2015 period in the Czech lands. Here, the authorship team expand their analysis of this decade to the wider Central European region, and convincingly demonstrate its climatological and societal significance using documentary data and gridded climate reconstructions. My comments and suggestions are therefore of a minor nature.

Specific comments:

Section 3 - I think it would be helpful if the maps in the supplementary material were included here (if space permits) or at least referred to in the first sentence of the methods section. This would help the reader place the results described in Section 4.

p. 6 lines 3-8 - here it is claimed that "documentary-based reconstructions were considered in the light of the corresponding European maps..."; however in my view the paper seems to lack an explicit comparative discussion between the documentary data discussed in sections 4.1.1 to 4.1.10 and the maps, which are merely referred to in the subheadings. It would be helpful to the reader offer a more blended discussion of the selected written reports and the extent/severity of precipitation shown in the maps in question, given that the integration of these forms of data is a - if not the - unique aspect of the paper.

p. 14 lines 27-29 - I would suggest adding clarification/interpretation here on how the weaker uniqueness of the 1531-1540 decade in the OWDA can be explained. Is this, as the authors later write in the conclusion, due to the two reconstructions representing different hydroclimatic variables?

p. 14 lines 10-11 - it is stated that there are no indications that the more severe impacts were of multi-country or multi-year natures. Can the authors offer any thoughts on why this may have been? For example, could this be partly because individual dry years in this decade were generally broken by normal or wetter years rather than a prevalence of back-to-back drought events? I do not doubt that this question could be the topic of an entire paper, however some expansion on this point would add a greater interpretive element to 'impacts' sections.

Technical corrections:

p. 1 line 21 - remove 'the' before 'documentary evidence'

p. 2 lines 31-33 - suggest rewriting in an active voice, e.g. 'This study aims to utilise and analyse...'

p. 5 lines 7-13 - it would read better if points (i) and (iii) were given a header (e.g.

Drought indices) to match the points that follow, rather than starting with a citation.

p. 18 line 2 - check that the end of this sentence reads ok.

---

## Referee Comment (RC2) · Anonymous Referee #2 · 18 Sep 2020

This is an exemplary paper that uses a very large amount of documentary and other proxy evidence for a 'deep dive' into one particular decade in the sixteenth century. There is clearly a huge amount of work that has gone into compiling these records, not to mention the work that went into putting them together in the first place.

I therefore only have two very minor suggestions:

I would like to know a bit more about the discrepancy with the Luterbacher et al. (2002) NAO reconstruction. This study uses loads of data, so does this suggest a limitation in the Luterbacher et al. paper?

Can you say which decades are drier in the OWDA record, and give some suggestion of why there is a discrepancy? This is particularly important given the conclusion that

the decade was the driest, (rather than saying that it may have been the driest). If you're going to be that confident you need to say why you're effectively discounting the OWDA record.

Otherwise great, thanks for compiling these records!

―――――――――――――――――

---

## Author Comment (AC1) · 21 Sep 2020

General comments: This is an excellent paper that brings together a range of data sources and datasets in order to analyse the characteristics and significance of the 1530s, which the authors have previously shown to be the driest summer decade of the period 1501-2015 period in the Czech lands. Here, the authorship team expand their analysis of this decade to the wider Central European region, and convincingly demonstrate its climatological and societal significance using documentary data and gridded climate reconstructions. My comments and suggestions are therefore of a minor nature. RESPONSE: We would like to thank the anonymous referee #1 for generally positive evaluation of our paper and very useful critical comments below, which we are trying to include into manuscript with the aim to improve the recent paper.

Specific comments: Section 3 - I think it would be helpful if the maps in the supplementary material were included here (if space permits) or at least referred to in the first sentence of the methods section. This would help the reader place the results described in Section 4. RESPONSE: Accepted. Because we specify position of every individual location in the first use in the manuscript with respect to corresponding country, we would rather prefer to preserve two corresponding maps in the supplementary material. We corrected the corresponding sentence in the first paragraph of Methods as follows: "Historical place names for the locations to which the report or event described was attributed were updated to their recent equivalents and their exact positions were expressed in Figure S1 in Supplement."

p. 6 lines 3-8 - here it is claimed that "documentary-based reconstructions were considered in the light of the corresponding European maps. . ."; however in my view the paper seems to lack an explicit comparative discussion between the documentary data discussed in sections 4.1.1 to 4.1.10 and the maps, which are merely referred to in the subheadings. It would be helpful to the reader offer a more blended discussion of the selected written reports and the extent/severity of precipitation shown in the maps in question, given that the integration of these forms of data is a - if not the - unique aspect of the paper. RESPONSE: There is nearly impossible to compare point documentary reports with gridded quantitative reconstructions of precipitation totals on the one hand and of drought expressed by scPDSI on the other. The aim to use these gridded maps is to show the spatial patterns of relatively different quantitative hydroclimatic information in the broader European scale. In description of individual years it represents rather some additional information to reported documentary evidence. Moreover, in Fig. 13 and related text to this figure in Section 4.3 we stressed quite important spatial differences between the both variables.

p. 14 lines 27-29 - I would suggest adding clarification/interpretation here on how the

weaker uniqueness of the 1531-1540 decade in the OWDA can be explained. Is this, as the authors later write in the conclusion, due to the two reconstructions representing different hydroclimatic variables? RESPONSE: Accepted. The following sentences were added: "This is probably related to the fact that summer hydroclimatic patterns are only one factor among others influencing a tree growth. Moreover, the OWDA, as a spatial reconstruction, is calculated from numerous tree-ring width series and is more spatially heterogeneous (see Fig. 13b). Thus, if we compare series of the Czech drought indices with the mean OWDA series over a relatively large window (5–25°E and 45–55°N), OWDA seems to be rather smoothed (compare dry Bohemia and eastern Germany with wetter south-east Poland and Austria in Fig. 13b)."

p. 14 lines 10-11 - it is stated that there are no indications that the more severe impacts were of multi-country or multi-year natures. Can the authors offer any thoughts on why this may have been? For example, could this be partly because individual dry years in this decade were generally broken by normal or wetter years rather than a prevalence of back-to-back drought events? I do not doubt that this question could be the topic of an entire paper, however some expansion on this point would add a greater interpretive element to 'impacts' sections. RESPONSE: The reviewer has probably in mind page 18 and the last sentence in point (i) in Conclusion. To fulfil this reviewer's request, we decided include the following paragraph at the end of Section 5.2 as follows: "As shown by described individual impacts in Section 4.1, it follows that there were no indications of multi-country or multi-year severe impacts. It can be related to several circumstances. Because of the differences in sensitivity of the various agricultural crops to temperatures and precipitation in different stages of their development, their yields were besides summer significantly influenced also by weather course in other seasons, particularly spring, but also autumn and winter. Further, dry years with possible severe impacts were broken by years with normal or wetter patterns, it is there was no longer concentration of worse weather patterns strongly influencing agricultural production on the long term in a negative way. Moreover, great spatial precipitation variability meant that not the whole central Europe was affected by the same way, and to

some extent trade could buffer the potential negative effects. Moreover, relatively stable socio-political situation without wars and extensive plague epidemics allowed administration to settle regionally limited shortages, despite the fact that some regions such as the Carpathian Basin generally suffered from Turkish and German attacks, taxation-related economic issues and internal political controversies (e.g. Perjés, 1989), which circumstances also make it more difficult to single out the problems primarily caused by unfavourable weather conditions. In this context we have to take in account that attention to recording of weather anomalies was sometimes of secondary importance compared to reporting other societal events, i.e. some impacts could remain unnoticed." Reference: Perjés, G.: The Fall of the Medieval Kingdom of Hungary: Mohacs 1526-Buda 1541, Atlantic Research and Publications, Boulder, Colorado, 1989.

Technical corrections: p. 1 line 21 - remove 'the' before 'documentary evidence' RESPONSE: Accepted and corrected as requested.

p. 2 lines 31-33 - suggest rewriting in an active voice, e.g. 'This study aims to utilise and analyse. . .' RESPONSE: Accepted and corrected as requested.

p. 5 lines 7-13 - it would read better if points (i) and (iii) were given a header (e.g. Drought indices) to match the points that follow, rather than starting with a citation. RESPONSE: Accepted. The corresponding sentences were corrected as follows: "(i) Seasonal series of three drought indices were calculated for what is now the Czech Republic in the 1501–2015 period (Brázdil et al., 2016a): Standardised Precipitation Index – SPI (McKee et al., 1993), Standardised Precipitation Evapotranspiration Index – SPEI (Vicente-Serrano et al., 2010) and Z-index (Palmer, 1965)." "(iii) Tree-ring widths were used to calculate gridded ($0.5°$ x $0.5°$) summer self-calibrated Palmer Drought Severity Index (scPDSI) for The Old World Drought Atlas (OWDA) (Cook et al., 2015), covering the 0–2012 CE period (http://drought.memphis.edu/OWDA/Default.aspx; accessed on 8 January 2020)."

p. 18 line 2 - check that the end of this sentence reads ok RESPONSE: Accepted. The

sentence was corrected as follows: "Reconstructions based on tree-ring widths tend to attribute the JJA scPDSI of 1531–1540 to the end of the first ten driest summers."

---

## Author Comment (AC2) · 21 Sep 2020

This is an exemplary paper that uses a very large amount of documentary and other proxy evidence for a 'deep dive' into one particular decade in the sixteenth century. There is clearly a huge amount of work that has gone into compiling these records, not to mention the work that went into putting them together in the first place. RESPONSE: We would like to thank the anonymous referee #2 for generally positive evaluation of our paper and two suggestions, which we are trying to explain below.

I therefore only have two very minor suggestions:

I would like to know a bit more about the discrepancy with the Luterbacher et al. (2002)

NAO reconstruction. This study uses loads of data, so does this suggest a limitation in the Luterbacher et al. paper? RESPONSE: We do not think that this finding has anything common with the limitation of the NAO reconstruction. The discrepancy may be related to the fact that NAO represents large-scale circulation pattern and its manifestation on of the Czech Lands territory may be complicated by other local/regional factors (air temperature, soil moisture, etc.). Moreover, both the Luterbacher et al. NAO and drought indices used in our paper as a primary data source are proxy reconstructions with some degree of uncertainty. This may be another reason for the explanation of the found differences not only with NAO but also with other proxy-based reconstructions used for comparison.

Can you say which decades are drier in the OWDA record, and give some suggestion of why there is a discrepancy? This is particularly important given the conclusion that the decade was the driest, (rather than saying that it may have been the driest). If you're going to be that confident you need to say why you're effectively discounting the OWDA record. RESPONSE: The following decades in OWDA were drier than 1531-1540: JJA PDSI: 1861–1870, 1831–1840, 1741–1750, 1941–1950, 1781–1790, 1511–1520, 1901–1910, 1631–1640 JJA DAI: 1861–1870, 1831–1840, 1941–1950, 1781–1790, 1741–1750, 1801–1810, 1511–1520 We hope that some response to your comment could be the new version of the corresponding paragraph which we complemented based on the comments of the referee #1: "The uniqueness of the 1531–1540 decade is weakened when decadal characteristics in the 1501–2012 period for the same window are calculated from OWDA (Cook et al., 2015). The 1531–1540 decade in terms of scPDSI for JJA emerges as the ninth driest, while JJA DAI is the eighth at a threshold of –1 (Fig. 15c,d). This is probably related to the fact that summer hydroclimatic patterns are only one factor among others influencing a tree growth. Moreover, the OWDA, as a spatial reconstruction, is calculated from numerous tree-ring width series and is more spatially heterogeneous (see Fig. 13b). Thus, if we compare series of the Czech drought indices with the mean OWDA series over a relatively large window (5–25°E and 45–55°N), OWDA seems to be rather smoothed (compare dry Bohemia

and eastern Germany with wetter south-east Poland and Austria in Fig. 13b)." Moreover, inspection of data sources used for OWDA spatial reconstruction reveals that for the territory of the Czech Lands only a single TRW chronology (fir - Abies alba), was included. This chronology explains only a part of the drought variability (so do drought indices used in our study) and is further combined with other data in spatial OWDA reconstruction.

Otherwise great, thanks for compiling these records! RESPONSE: Many thanks!